# Rattle drum-inspired triboelectric nanogenerator with enhanced output using charge dispatch and magnetic repulsion pendulum

Wei Tang [1,2], Hongfang Li[1,2], Jiawei Li[1,2], Weiyu Zhou[1,2], Jiaqi Duan[1,2], Yongsheng Wen[1,2], Lingyu Wan [1,2] ✉ & Guanlin Liu [1,2] ✉

Densifying triboelectric layers benefits triboelectric nanogenerators (TENGs) by increasing output, improving spatial utilization, and reducing costs. However, structural densification imposes limitations that hinder further output enhancement. A charge dispatch strategy is developed in our proposed rattle drum inspired TENG, which mitigates charge cancellation via path diversion and alleviates electrostatic shielding through mode transformation, yielding over 6x output versus traditional models. Further structural designs to improve layer contact-separation efficiency, including laser etching and contact push pins, raise the triboelectric surface density to 2.76 cm⁻¹. A comprehensive framework is established, encompassing theoretical modeling, engineering optimization, and extensive experimental validation. Furthermore, the generator can capture weak wave energy when equipped with a magnetic repulsion pendulum, boosting motion amplitude and output by 558% and 1662%, respectively, demonstrating scenario adaptability expansion. Here, we show strategies to further elevate layer density and pathways to enhance TENG output.

The evolution of energy utilization forms and technologies has profoundly shaped human civilization. The early use of firewood combustion enabled the conversion of chemical energy into heat and light, extending human lifespan and activity time, and facilitating the transition from primitive to agricultural societies. Subsequent inventions such as plows, waterwheels, and windmills exploited animal, hydraulic, and wind energy to enhance agricultural productivity, promote population growth and urbanization, and establish agricultural civilization. The Industrial Revolution was propelled by steam engines, which transformed coal's chemical energy into mechanical power, driving the development of modern industry. The advent of electromagnetic generators further enabled the large-scale conversion of mechanical energy, sourced from wind, thermal, hydro, or nuclear energy, into electricity, liberating energy use from geographical constraints. This transformation laid the foundation for the electrified era, where electricity became the core infrastructure of modern society and catalyzed widespread industrial and social progress. In the era of the Internet of Things (IoT) and smart cities, billions of electrical devices and smart sensors are now embedded throughout human society[1,2]. Traditional power supply methods, such as power lines and batteries, are increasingly insufficient to support the widespread and low-power demands of IoT devices, highlighting the need for a paradigm shift toward advanced energy harvesting solutions. One promising approach to converting ambient environmental energy into

[1]Center on Nanoenergy Research, Institute of Science and Technology for Carbon Peak & Neutrality, School of Physical Science & Technology, Guangxi University, Nanning, China. [2]State Key Laboratory of Featured Metal Materials and Life-cycle Safety for Composite Structures, Guangxi University, Nanning, China. ✉e-mail: lyw2017@gxu.edu.cn; guanlinliu@gxu.edu.cn

electricity is triboelectric nanogenerators (TENGs)[3]. Harnessing environmental mechanical energy through TENGs has emerged as a key research focus, driven by their high-entropy energy conversion efficiency, lightweight design, and ease of fabrication[2,4–6]. Over the past decade, TENGs have achieved significant breakthroughs in micro-nano energy harvesting[7,8], self-powered sensing[9,10], high-voltage direct-drive power supplies[11,12], and blue energy[13,14], gradually advancing toward commercialization and playing an increasingly vital role in the global energy transition.

However, the path to the commercialization of TENGs is fraught with stumbling blocks, including suboptimal output performance, limited mechanical lifespan, high fabrication costs, and limited adaptability to various environments[15,16]. In particular, for harvesting similar mechanical energies, such as micro-wind and low-frequency wave energy sources, various TENG structures and mechanisms have been proposed[17–22]. However, considerable debate remains over the most efficient TENG structure for energy harvesting, with researchers still divided and unable to reach a consensus. This disagreement stems not only from the significant variations in practical applications but also from the lack of a universally accepted, straightforward metric for evaluating the performance of different TENG structures[23–27]. The introduction of triboelectric surface density (TSD) has, to some degree, provided a resolution to these controversies[16]. TSD is defined as the ratio of the surface area of the triboelectric layer to the total volume of the TENG device. Considering that the triboelectric layer is central to TENG operation, its surface area is a critical factor influencing output performance[28,29]. As such, TSD can serve as a direct or indirect measure of the triboelectric layer density, spatial utilization, overall output performance, and even fabrication costs in TENG design. In addition, when combined with the specific surface area (SSA)[16], the triboelectric surface area per unit mass, it enables evaluation of the device's lightweight design. Together, TSD and SSA constitute key structural performance metrics for TENG devices. Improving TSD is expected to overcome current technical bottlenecks and accelerate TENG industrialization. Compactness, triboelectric layer integration, and lightweight design are regarded as future trends for distributed energy devices[16]. However, as TSD and triboelectric layer density increase, achieving a sufficient contact-separation distance becomes increasingly challenging. Furthermore, in traditional free-standing vertical contact-separation (FCS) structures, the decomposed fundamental unit is the single-electrode mode, which inherently suffers from electrostatic shielding effects[30]; resulting in device internal electrostatic shielding intensifying with increasing layer density, and the use of shared electrode layers leads to severe charge cancellation between stacked units, further limiting the overall electrical output. Therefore, designing efficient charge transmission and dispatch channels within a high-density triboelectric layer to tackle these bottlenecks, while simultaneously expanding TENG applicability to diverse energy source scenarios, is critical to overcoming these technological barriers and advancing TENG commercialization.

In this work, we introduce a rattle drum-inspired triboelectric nanogenerator (RD-TENG), achieving charge dispatch in a high-density triboelectric layer. The integration of three-electrode technology and alternating film coatings allows free electrons to shuttle between the internal electrodes, realizing microscale charge diversion and macroscale mode transformation of the basic working units. This design effectively mitigates the electrostatic shielding and charge cancellation effects, leading to a high output at the external electrode. Combining structural innovations based on this strategy to improve contact-separation efficiency, yielding a TSD of 2.76 cm$^{-1}$ for the RD-TENG. The integrated vibrating steel sheet, etched using laser engraving technology, replaces traditional elastic auxiliaries with elastic cantilevers. The electrode connections employ a contact push-pin electrode design, removing the need for complex wire

connections, mechanical fasteners, adhesives, and welding. A latch-fixed looping method is used for rapid, layer-by-layer stacking, simplifying the fixing and stacking of power generation units while ensuring stability. Furthermore, using a rotating-clasp device enables the RD-TENG to be flexibly assembled into arrays in a Lego-like modular fashion, providing technical support for large-scale deployment and integration. These innovations significantly improve spatial utilization, output performance, and fabrication efficiency, reduce production costs, and enhance the overall stability of individual devices and arrays. Benefiting from the contact-separation mode, the mechanical lifespan of the device is ensured. Under both theoretical and structural designs, the RD-TENG generates a peak voltage of 2200 V and a volumetric peak power density (VPD) of 136.74 W m$^{-3}$, effectively harvesting mechanical energy to power electronic devices. Compared to similar devices, the five-in-one evaluation system, including structural and electrical performance indicators, in aggregate, the performance exceeds that of previous studies (Table 1 and Supplementary Note 6)[31–45]. To expand the RD-TENG's adaptability to various scenarios, including broadband, low-amplitude, and variable marine wave energy harvesting[46,47], we designed a frequency-reducing and amplitude-amplifying magnetic repulsion pendulum (MRP), converting high-torque, low-amplitude water wave motion into low-torque, high-amplitude swing motion. A minimal external stimulus is sufficient to trigger the swinging of the MRP, thereby driving the RD-TENG to operate efficiently. Under identical excitation conditions, the motion amplitude and output increase by 558% and 1662%, respectively. Field-based marine wave energy harvesting was successfully demonstrated, with the device effectively generating power to supply electronic devices even in gentle sea conditions after low tide. Additionally, the introduction of orthogonal experimental methods[48,49] enables systematic exploration of the RD-TENG's response to various excitation parameters, such as vibration and wave energy, while introducing new technologies and experimental methodologies to the TENG field.

## Results
### Mechanism and theoretical model of charge dispatch strategy
As shown in Supplementary Note 1, the traditional FCS-TENG models suffer from limited output when the triboelectric layer is densified, due to electrostatic shielding, charge cancellation, and reduced interlayer contact-separation efficiency. Inspired by the rattle drum (Supplementary Note 2), we replace the pendulums, pendulum line, and drum surfaces with corresponding TENG material components (Fig. 1a, Supplementary Fig. 4b, and Supplementary Note 2). By continuously alternating between charge dispatch and field equilibrium, the pendulum enables charge migration and compensation through its connecting channel during motion, achieving a dynamic balance of electric potential. This results in an alternating balanced potential field between the two pendulums (golden), which, under the action of the two pendulums, generates a high electric potential difference in the external electrode pair of the TENG (red and green), facilitating charge transfer (Fig. 1a[i–ii]). Based on this concept, we designed a rattle drum-inspired TENG (RD-TENG) and developed charge dispatch strategies. As shown in Fig. 1b and Supplementary Notes 2 and 3, we proposed a three-electrode system that is fundamentally distinct from conventional three-electrode configurations (comparative analysis in Supplementary Note 4)[50–53]. When vibrating electrode 2 is stimulated by external mechanical energy, causing all oscillators to move to the right (Fig. 1b[i–ii]), the non-coated vibrating electrode 2 oscillator contacts the left-coated fixing electrode 3, and the full-coated oscillator contacts the right-coated fixing electrode 1. Electrostatic induction causes the non-coated vibrating electrode 2 to induce positive charges, and electrons are dispatched along the internal vibrating electrode 2 channel from the non-coated sheet to the full-coated sheet, achieving charge compensation on vibrating electrode 2. At this point,

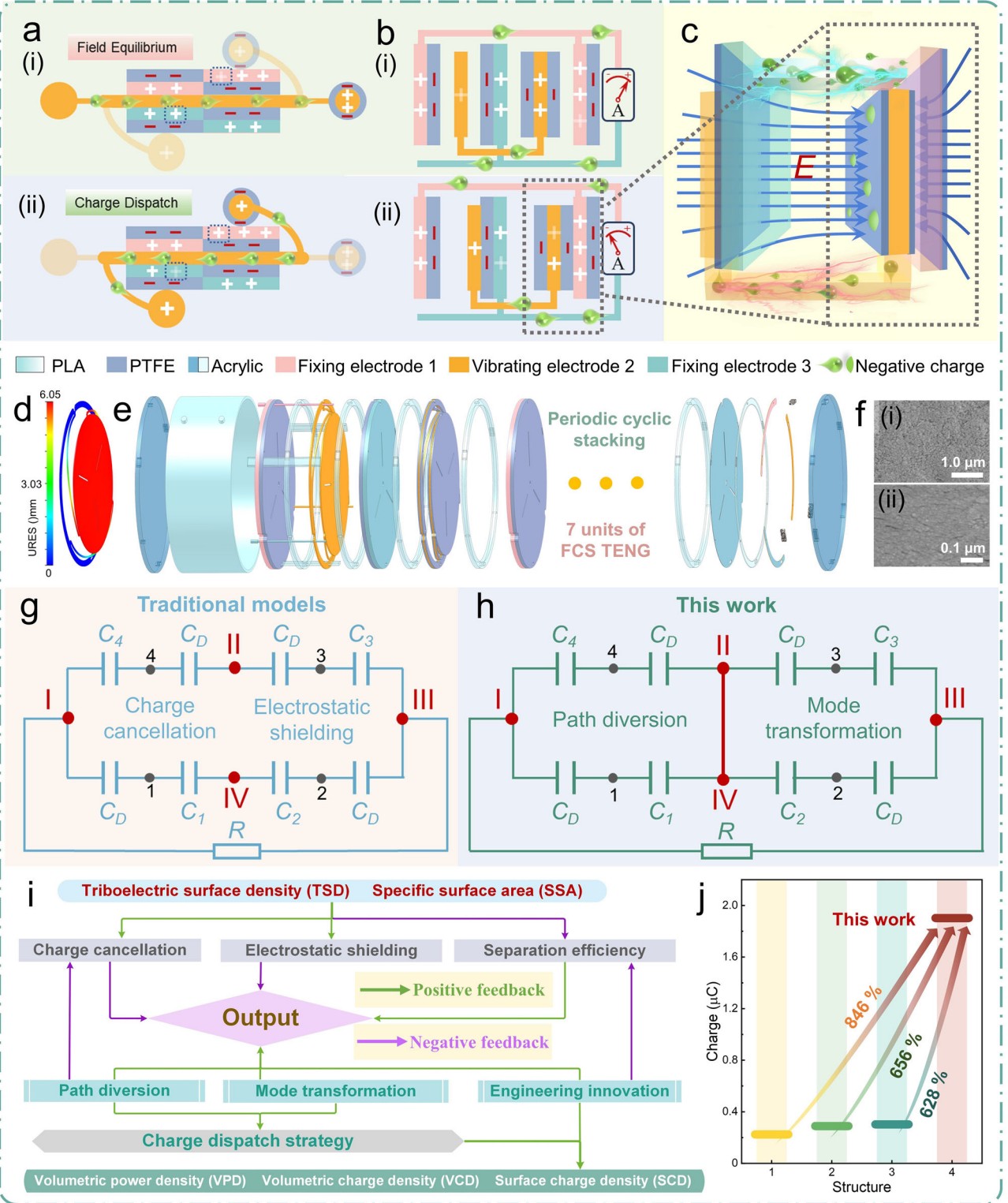

**Fig. 1 | Mechanism of the rattle drum inspired TENG (RD-TENG). a** Charge dispatching principle of the rattle-drum-type TENG. **b** RD-TENG model and its charge dispatching principle (a structural periodic in the stacked layer, the electrode is grounded). **c** Effect of the electric field from vibrating electrode 2 on the external electrode pair, $E$ denotes the electrostatically superposed field of the oscillator. **d** Strain simulation of the vibrating sheet under 1 N stress. **e** Schematic diagram of the RD-TENG device. **f** SEM characterization images: (i) Spring steel sheet and (ii) PTFE. Equivalent capacitance circuit model, C capacitance, R load, and dots are circuit nodes: **g** traditional models, **h** this work. **i** Correlation mapping between structural indicators and electrical performance metrics. Positive feedback: one quantity enhances another; negative feedback: one quantity suppresses another. **j** Comparison of the output of the RD-TENG with traditional models. Source data are provided as a Source Data file.

the negative charge on the full-coated vibrating electrode 2's PTFE film exceeds the positive charge on the steel sheet, emitting a net electrostatic field. This field acts on fixing electrode 1, inducing positive charges on it, and the negative charge travels through the 1–3 electrode channel to fixing electrode 3 to compensate for the potential field effect caused by the non-coated vibrating electrode 2. This forms another charge output channel opposite to the charge-balancing channel of electrode 2. Microscopically, the internal short-circuit electrode enables charge path diversion, preventing cancellation from reverse flows in shared paths. Macroscopically, it transforms each external electrode from single-electrode mode into contact-separation mode by pairing it with vibrating short-circuit electrodes, enabling mode transformation and resolving the electrostatic shielding (Supplementary Notes 3 and 5). As shown in Fig. 1c, when the full-coated vibrating electrode 2 contacts the fixing electrode 1, its electric field triggers the charge generation and electrostatic induction effects at the fixing electrode 1 interface, causing the external charge output channel to transfer more charge. The corresponding potential distribution is simulated in Supplementary Fig. 18a. As vibrating electrode 2 moves, the 1–3 electrode pair exhibits significant potential variations. The dynamic potential trends of each electrode are presented in Supplementary Fig. 18b, showing the potential evolution as vibrating electrode 2 transitions from the vicinity of fixing electrode 1 to fixing electrode 3. Sheets 2 and 4, corresponding to the non-coated and full-coated vibrating electrode 2, are short-circuited, resulting in identical and synchronous potential variations. Meanwhile, fixing electrode 3 experiences a more pronounced potential fluctuation due to the influence of vibrating electrode 2, leading to an increased potential difference in the 1–3 electrode pair. In the RD-TENG model, three electrodes can form alternating current (AC) electrode pairs in different combinations or be rectified together to form a direct current (DC) electrode pair (Supplementary Fig. 19). The output configurations for the electrode pairs are schematically illustrated in Supplementary Fig. 20.

Based on the above RD-TENG model, we fabricated an integrated elastic vibrating steel sheet (Supplementary Fig. 21). It is laser-etched into three parts: the outer fixed ring, the inner oscillator, and the cantilever bridge connecting the two parts. The elastic cantilever replaces the traditional elastic auxiliary components in the TENG device, and the oscillator's weight replaces the conventional counterweight, optimizing the spatial layout. When the vibrating sheet is excited by an external force, the oscillator, connected by the cantilever bridge, can perform a reciprocating motion like a simple harmonic spring oscillator system (Supplementary Fig. 22). Figure 1d simulates the strain situation of the vibrating sheet under a 1 N external force. The displacement of the oscillator on one side can reach 6.05 mm, and the three cantilevers show good elastic response, ensuring that the oscillator responds well under periodic excitation. As shown in Fig. 1e, fixed sheets are placed on both sides of the vibrating sheet, ensuring contact separation to form an RD-TENG device. An annular acrylic ring gasket separates the vibrating sheet and the fixed sheets, ensuring sufficient vertical separation space between the oscillator and the left and right fixed sheets. The full-model view and engineering layout are shown in Supplementary Fig. 23. After cycling through 7 FCS-TENG units, a high-density layer device consisting of 14 vertically contact-separated units, as shown in Supplementary Fig. 24, is formed. Figure 1f(i–ii) displays the microstructure of the steel sheet and PTFE film surface. The microstructure between the friction materials promotes the contact electrification efficiency during the contact-separation process.

The RD-TENG employs a charge dispatch strategy that, at the macroscopic level, transforms the fundamental unit from a single-electrode mode (Supplementary Fig. 3 in Supplementary Note 1) to a vertical contact-separation mode (Supplementary Fig. 7 in Supplementary Note 5), thereby resolving electrostatic shielding. While at the

microscopic level, it is characterized by the use of charge path diversion to overcome the challenge of charge cancellation caused by shared electrodes. The corresponding equivalent capacitance circuit transformation is shown in Fig. 1g, h and Supplementary Note 5. Introducing the short-circuit electrode pair (node II-II, Fig. 1h) transitions the single-loop charge path into a dual-loop path configuration, facilitating charge diversion and overcoming electrostatic shielding and charge cancellation. For traditional models, the short-circuit transferred charge $Q$ and open-circuit voltage $V$ are given by:

$$Q = 0 \tag{1}$$

$$V = 0 \tag{2}$$

For the RD-TENG:

$$Q = \frac{\frac{\sigma_T S}{C_D}\left(\frac{1}{C_{s-x}} - \frac{1}{C_x}\right)}{\left(\frac{1}{C_x} + \frac{1}{C_D}\right)\left(\frac{1}{C_{s-x}} + \frac{1}{C_D}\right)} \tag{3}$$

$$V = \frac{2\sigma_T S}{C_D} \frac{\frac{1}{C_x} - \frac{1}{C_{s-x}}}{\frac{1}{C_x} + \frac{1}{C_{s-x}} + \frac{2}{C_D}} \tag{4}$$

Where the steel-sheet thickness is negligible compared to the electrode spacing; $R = 0$ under short-circuit conditions; $C_D$ denotes dielectric (parasitic) capacitance; $\varepsilon_0$ is the vacuum permittivity; $\varepsilon_r$ the dielectric relative permittivity; $S$ the electrode area; $x$ the gap from the leftmost dielectric film to the left electrode (gap of capacitor $C_1$); $s$ the spacing between the two dielectric layers; $s - x$ the gap from the right electrode to the right dielectric film (gap of capacitor $C_2$); and $d$ the dielectric thickness, and:

$$C_x = \frac{\varepsilon_0 S}{x} \tag{5}$$

$$C_{s-x} = \frac{\varepsilon_0 S}{s - x} \tag{6}$$

$$C_D = \frac{\varepsilon_0 \varepsilon_r S}{d} \tag{7}$$

Thus, we obtain a quantitative theoretical model for the RD-TENG, demonstrating that the equivalent capacitance configuration effectively overcomes the lack of electrical output encountered in traditional models. Detailed analysis, derivations, and visualizations are presented in Supplementary Note 5.

To rigorously evaluate the RD-TENG's performance, we established a comprehensive assessment framework (Table 1 and Supplementary Note 6), which includes structural metrics such as TSD and SSA, as well as electrical performance metrics such as VPD, volumetric charge density (VCD), and surface charge density (SCD). The correlation among these metrics is illustrated in Fig. 1i. Under the charge dispatch strategy and structural innovations, improvements in structural metrics directly enhance electrical performance. To better clarify the device performance, additional metrics for array modularity, scenario adaptability, and durability were introduced, together forming the comprehensive evaluation framework of the RD-TENG. Compared to the conventional direct FCS-TENG dense stacking model without charge dispatch strategies, the RD-TENG generates outputs that are 846%, 656%, and 628% higher than those of traditional structures 1, 2, and 3, respectively (Fig. 1j). As shown in Table 1, the four electrode pairs of the RD-TENG exhibit VPD outputs of 136.74 W m⁻³, 102.51 W m⁻³, 84.97 W m⁻³, and 114.00 W m⁻³, respectively. These outputs are

**Table 1 | Comparison of the RD-TENG with other works in terms of various parameters and indicators**

| Type | Reference | Charge (μC) | Friction area (cm²) | Volume bulk (cm³) | Weight (g) | SCD (μC m⁻²) | VCD (mC m⁻³) | TSD (cm⁻¹) | SSA (cm² g⁻¹) | VPD (W m⁻³) |
|---|---|---|---|---|---|---|---|---|---|---|
| RD-TENG | This work | 4.22 | 999.35 | 361.73 | 345 | 42.23 | 11.69 | 2.76 | 2.9 | 1–3 electrode 136.74 1–2 electrode 102.51 2–3 electrode 84.97 Rectified electrode 114 |
| FH-TENG | Ref. 31 2024 | 7.9 | 2052.75 | 769.69 | 621.2 | 38.48 | 10.26 | 2.67 | 3.3 | 14.94 |
| OM-TENG | Ref. 32 2024 | 60.82 | 15,226.63 | 8651.49 | 7110 | 39.94 | 7.03 | 1.76 | 2.14 | 28.9 |
| SO-TENG | Ref. 33 2024 | 1.5 | 900 | 1359.78 | 800 | 16.67 | 1.1 | 0.66 | 1.13 | 4.44 |
| D-Z-TENG | Ref. 34 2024 | 1.2 | 360 | 267 | 500 | 33.33 | 4.49 | 1.35 | 0.72 | 55.4 |
| DA-TENG | Ref. 35 2024 | 0.62 | 343 | / | / | 18.08 | 0.19 | 0.8 | / | 7.51 |
| GA-TENG | Ref. 36 2024 | 1.8 | 343 | 147 | / | / | 0.122 | 2.3 | / | 20.4 |
| LI-TENGs | Ref. 37 2023 | 0.76 | 216 | 108 | 115.6 | 35.19 | 7 | 2 | 1.87 | 48.47 |
| T-TENG | Ref. 38 2023 | 1.15 | 460.86 | 430.71 | 358 | 24.95 | 2.67 | 1.07 | 1.29 | 18.9 |
| HM-TENG | Ref. 39 2022 | 2,4 | 951.72 | 924 | 742.6 | 25.22 | 2.6 | 1.03 | 1.28 | 2.44 |
| O-TENG | Ref. 40 2022 | 29.93 | 12,160 | 6170.73 | 7300 | 24.61 | 4.85 | 1.97 | 1.67 | 1.62 |
| MH-TENG | Ref. 41 2023 | 2.9 | 783.78 | 698.28 | 353.25 | 37.00 | 4.15 | 1.12 | 2.2 | 23.2 |
| D-TENG | Ref. 42 2022 | 0.87 | 480 | 5400 | 280.8 | 18.13 | 0.16 | 0.09 | 1.71 | 6.35 |
| S-TENG | Ref. 43 2022 | 0.16 | 62.8 | 448 | 121.1 | 25.48 | 0.36 | 0.14 | 0.52 | 7.39 |
| F-TENG | Ref. 44 2022 | 0.18 | 65 | 2574.26 | 80.96 | 27.69 | 0.07 | 0.03 | 0.8 | 16.96 |
| SR-TENG | Ref. 45 2022 | 0.15 | 202.3 | 736.63 | 281 | 7.41 | 0.2 | 0.27 | 0.72 | 15.4 |

*TSD* triboelectric surface density, *SSA* specific surface area, *VPD* volumetric power density, *VCD* volumetric charge density, *SCD* surface charge density.

significantly higher than those of similar previous works, demonstrating the excellent mech-elect conversion capability. The RD-TENG structure achieves high spatial utilization with the TSD reaching 2.76 cm⁻¹, SSA reaching 2.9 cm² g⁻¹, and after tuning by the MRP, the VCD reaches 11.69 mC m⁻³, SCD reaches 42.23 μC m⁻². Compared with those of similar structures reported previously, these structural parameters demonstrate the RD-TENG's superiority in terms of output characteristics and structural performance.

**Structural designs**

Based on the aforementioned charge dispatch principle and model, we have implemented a series of structural innovations for the RD-TENG. Figure 2a(i) shows one unit structural cycle cross-section image of the RD-TENG (for more details, see Supplementary Fig. 25), which consists of: two fixing sheets (Supplementary Fig. 26), two vibrating sheets, and acrylic ring gaskets arranged between each pair of steel sheets (Supplementary Fig. 27). Figure 2a(ii) and Supplementary Fig. 28 illustrate the left-side sectional view of the device, showing the staggered arrangement of three push pin holes and three latch holes arranged at 60° intervals along the circular periphery. Conduction between electrodes is facilitated through push pin contact conduction design (Supplementary Fig. 29), where each steel sheet is independently connected via the push pin. As shown in Fig. 2a(iii), Small apertures on the steel sheets ensure electrical contact with the push pin for conduction, while large apertures prevent conduction in non-conducting sheets. During assembly, conduction and non-conduction are managed by rotating and aligning the corresponding holes in the vibrating sheets, fixing sheets, and gaskets. This enhances the reliability of conductivity, eliminates the risk of short circuits, and reduces space occupation associated with traditional wiring, thus improving the contact-separation efficiency of the stacked layers. The generator units are fixed together using latches (Supplementary Fig. 29) and stacked with the looping method. As shown in Fig. 2a(iv), the working principles diagram illustrates the coordination between a single latch and

hole positions. Each sheet layer is sequentially fitted onto the latch through small holes, achieving layer-by-layer stacking and fixation of the device body, and eliminating the need for additional support frameworks and complex adhesive processes, while reducing time costs. These two methods enable a dense stacking arrangement with TSD reaching 2.76 cm⁻¹ and enhanced interlayer contact-separation efficiency.

The starting end of each push pin is designed with a 5 mm length and a larger outer diameter, ensuring a secure fit into the bottom sealing plate (Supplementary Fig. 30) and extending 3 mm, thereby serving as the lead-out terminal for the three electrodes (Supplementary Fig. 31). After assembling all the unit layers, a 0.5 mm thick PET ring gasket (the parameters are the same as Supplementary Fig. 30) is placed to insulate the final steel sheet's bare steel portion. Independent conduction is then achieved through the electrode conductive sheet (Supplementary Fig. 32) and the corresponding push pin. The packaging barrel (Supplementary Fig. 33) and top sealing plate (Supplementary Fig. 34) are fitted, with large holes at the electrode positions along the circular edge of the top sealing plate to insert conductive foam (Supplementary Fig. 35), facilitating independent conduction between the push pin, electrode sheet, and conductive foam. At this stage, a fully independent RD-TENG is fabricated (Supplementary Fig. 36), enabled by the charge dispatch strategy and structural innovations, and establishes a feasible TENG mode—three-electrode stacked FCS-TENG. The discussion on the scalability of each strategy can be found in Chapter 2 of Supplementary Note. 8. As shown in Fig. 2a(v), the device array design requires a rotating-clasp device (Supplementary Fig. 37), which allows for the flexible assembly of RD-TENG devices into an array. The channels reserved on the rotating-clasp device align with the four protrusions on the RD-TENG casing, and locking connections are achieved by inserting and rotating the protrusions into the channels. (Supplementary Fig. 38a and Movie 1), ensuring stable conduction between the electrodes and overall array stability. The connection method of push-pin electrodes follows a 1–1,

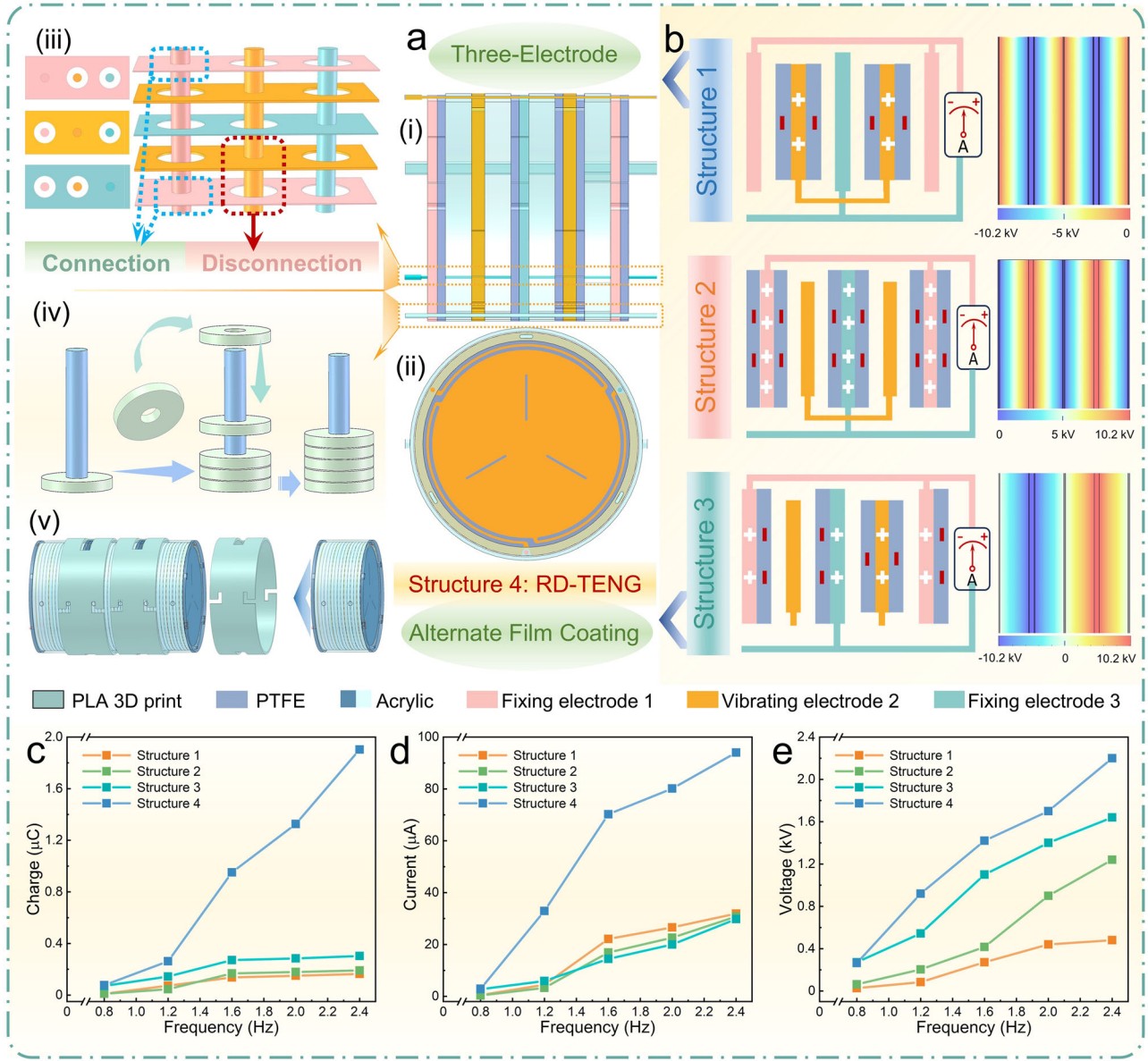

**Fig. 2 | Structural innovations of the RD-TENG. a** Sectional view of the device: (i) equivalent cross-sectional view of one structural cycle, ii) push pin and latch positions. The three structural innovations: (iii) Contact push pin electrode conduction technology, (iv) looping method for quick layer-by-layer stacking of power generation units, and (v) rotating-clasp device array technology. **b** Three dense stacking models of traditional FCS structures and their corresponding potential simulation diagrams (the electrode is grounded). **c–e** Comparison of charge, voltage, and current outputs at varying frequencies with a 70 mm amplitude between the RD-TENG 1–3 electrode pair and traditional models. Source data are provided as a Source Data file.

2–2, 3–3 correspondence (Supplementary Fig. 38b). Specifically, the push-pin head extending from the bottom of the second device is inserted into the conductive foam hole reserved at the top of the first device. The first device is then connected to the second device via the push pin, electrode sheet, and conductive foam, enabling parallel conduction. This assembly method enables devices to be arranged in large-area arrays, enhancing mechanical energy harvesting and advancing toward large-scale integration and commercialization.

To further verify the output differences between RD-TENG (Structure 4) and traditional models (Supplementary Note. 1), we conducted a comparative study. As shown in Fig. 2b, the potential simulation diagrams of the three structures show that electrodes 1 and 3 always maintain equal potential, and electrode 2 cannot achieve charge migration and compensation due to the lack of charge dispatch channels and the influence of the electrostatic shielding effect. As a result, no changing potential field is generated inside electrode 2, and

therefore, the potential between electrodes 1 and 3 does not undergo a significant change, leading to very low output. In contrast, RD-TENG combines alternate film coating and a three-electrode design. The internal electrode 2, formed by the short-circuit connection between the coated and uncoated vibrating sheet, achieves charge dispatch and generates a varying electric field, thereby enhancing the output of the external electrodes. This design effectively combines the three-electrode design of Structures 1 and 2 with the alternate film coating design of Structure 3. The same principle applies to the 1–2, 2–3, and rectifier electrode pairs. To compare the actual output differences, we performed output tests on the four electrode pairs of the four structures. The test platform is driven by a linear motor (Supplementary Fig. 39), and the output of each electrode pair is tested as a function of frequency at an amplitude of 70 mm. As shown in Figs. 2c–e, the short-circuit transfer charge, short-circuit current, and open-circuit voltage of the 1–3 electrode pair of Structure 4 are much higher than those of

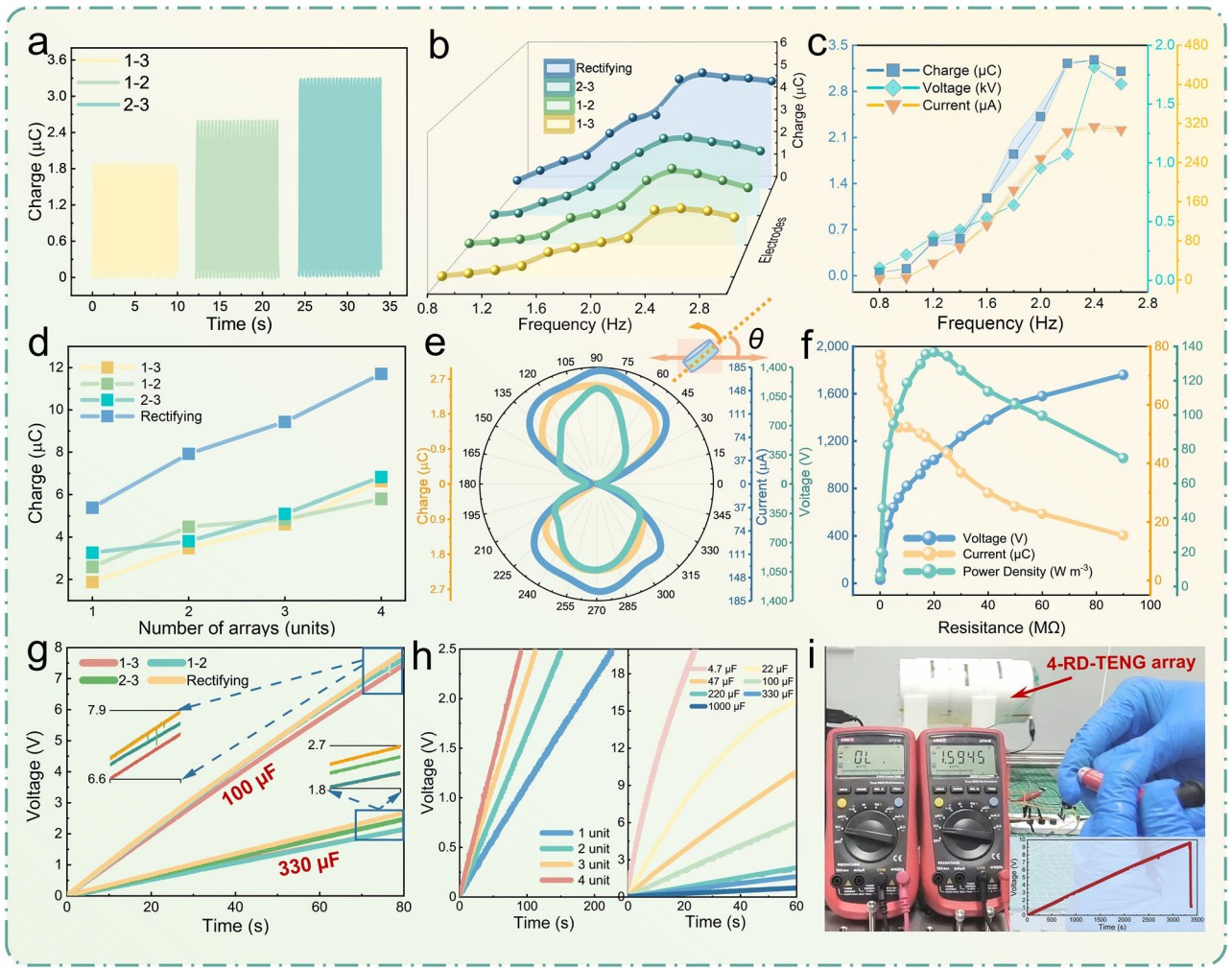

**Fig. 3 | RD-TENG output characteristics and applications for vibration energy harvesting. a** Output charge of each AC electrode pair. **b** Charge output of each electrode pair under varying frequencies at a 70 mm amplitude. **c** Charge, current, and voltage output of the 2–3 electrode pair under varying frequencies at a 70 mm amplitude. Data are presented as mean values ± SD (sample size $n_{Charge}$ = 15, $n_{Voltage}$ = $n_{Current}$ = 4). **d** Charge output of each electrode pair as the RD-TENG device array number increases. **e** Output characteristics of the 1–2 electrode pair with varying excitation azimuth angles. **f** Output voltage, current, and peak power density under different external loads. **g** Charging curves for 100 μF and 330 μF capacitors of each electrode pair. **h** Left: charging curves for different device quantities with a 1 mF capacitor, right: charging curves for different capacitors with the 2–3 electrode pair. **i** The rectified electrode pair of four RD-TENG devices array charges a 6.8 mF capacitor to drive two parallel multimeters with a rated voltage of 9 V. One multimeter is used to test the voltage of a 1.5 V dry battery. Source data are provided as a Source Data file.

Structures 1, 2, and 3. The charge output is 846%, 656%, and 628% higher than those of Structures 1, 2, and 3, respectively. The outputs of other electrode pairs are shown in Supplementary Table 1 and Fig. 40. The above qualitative simulation results and experimental measurements are consistent with the calculations derived from our constructed equivalent capacitance circuit model. It is worth noting that although Structures 1, 2, and 3 exhibit observable charge transfer outputs, the corresponding quantitative model predicts zero output. This discrepancy arises from the non-ideal conditions during actual testing, where experimental errors account for the deviation (Supplementary Note. 5). Structure 4 achieves a significant output improvement by short-circuiting all the layers of electrode 2 for charge dispatch, fully verifying the effectiveness and superiority of the charge dispatch strategy.

## Output characteristics under vibration energy drive

To investigate the output characteristics of RD-TENG under external mechanical energies, such as motion and vibration energy, we utilized the aforementioned linear motor platform to systematically simulate vibration energy and conduct testing. The output frequency of the linear motor is regulated by acceleration, amplitude, and end velocity, which correspond to the characteristics of most mechanical motion or vibration phenomena in nature. As shown in the Supplementary Note. 7, Experiment 1, the device output increases with frequency and amplitude, reaching saturation at the optimal combination of near 2.4 Hz frequency and 70 mm amplitude. Under this optimal combination, the output performance of the RD-TENG was tested. As shown in Fig. 3a, the short-circuit transferred charge of the 1–3, 1–2, and 2–3 electrode pairs is 1.9 μC, 2.6 μC, and 3.3 μC, respectively. The accumulated charge per cycle for the rectifier electrode pair is 5.54 μC (Supplementary Fig. 41a). The currents of the different electrode pairs are 94 μA, 183 μA, 260 μA, and 272 μA, and the voltages are 2200 V, 1400 V, 1220 V, and 1120 V, respectively (Supplementary Figs. 41b and 24c), reflecting the excellent electrical response and mechanical energy harvesting ability of RD-TENG.

To further validate the synchronicity of the output changes of the four electrode pairs in RD-TENG with varying external excitations, we focused on the short-circuit transferred charge as the primary metric.

We tested the output variations of the four electrode pairs concerning frequency changes at a 70 mm amplitude and with amplitude changes at a 2.4 Hz frequency (Fig. 3b and Supplementary Fig. 41d). The results show that the outputs of the four electrode pairs exhibit consistency with changing external excitations. The outputs increase or decrease synchronously with the excitation and reach their maximum values at a frequency of 2.4 Hz and an amplitude of 70 mm, consistent with the results of the orthogonal test analysis. To explore whether the other two output quantities are consistent with the changes in charge, we tested the 2–3 electrode pair over a frequency range from 0.8 Hz to 2.6 Hz. As shown in the results in Fig. 3c, at a 70 mm amplitude, when the frequency changes, the charge, current, and voltage exhibit synchronization with the frequency variation, reaching their maximum values at 2.4 Hz, consistent with the results obtained from the orthogonal test. The relationship of the 2–3 electrode pair with amplitude changes at a 2.4 Hz frequency also shows a strong correlation and synchronicity (Supplementary Fig. 41e), with all output quantities reaching their maximum values at an amplitude of 70 mm, further confirming the consistency with the orthogonal test results. Therefore, the three output quantities of the four electrode pairs in RD-TENG exhibit consistency with changes in external excitation. In subsequent parameter studies, testing any one electrode pair can be used to predict the output variations of the other electrode pairs.

To investigate the impact of different numbers of device arrays on output, we tested arrays consisting of 1–4 devices. The array assembly testing platform is shown in Supplementary Fig. 42a. As shown in Fig. 3d, the charge output increases with the number of devices under the optimal combination, while the variations in current and voltage are displayed in Supplementary Fig. 42b, c. The results indicate that both charge and current are positively correlated with the number of devices; as the number of devices increases, the output also increases, though at a diminishing rate. This phenomenon is attributed to the increase in overall weight as more devices are added, which results in greater inertia when driven by the linear motor. This causes tremors when the device reaches the end and reverses, affecting the synchronization of the vibrating sheets' movement (see the relevant analysis and proposed optimization strategies in Chapter 3 of Supplementary Note. 8). For an array of four devices, the single-cycle accumulated charge of the rectifier electrode pair reaches 11.70 μC, with the charge transfer for the 2–3 electrode pair at 6.83 μC, current at 529 μA, and voltage remaining essentially stable as the number of devices increases, proving that this array configuration is a parallel array. These results provide support for the large-scale integration of RD-TENG devices for collecting a broader range of mechanical energy.

Considering that mechanical or motion energy in nature typically exists in various directions, we further investigated the output characteristics of RD-TENG under changes in azimuth angle. As shown in Fig. 3e, the output variation of the 1–2 electrode pair of a single device was tested under optimal excitation conditions concerning angle, with the inset illustrating the angle relationship between the device and the linear motor's direction of motion. When the angle $\theta$ is 90° and 270°, the vibration direction of the vibrating sheet aligns with the external excitation direction, maximizing contact-separation efficiency and output; when the angle is 0° and 180°, the vibration direction is perpendicular to the excitation direction, resulting in the minimum output. At other angles, the output fluctuates between the maximum and minimum values. Supplementary Fig. 42d demonstrates the output pattern of the 1-2 electrode pair after the assembly of three devices, showing consistent output patterns for individual devices. In nature, mechanical linear motion exists not only in the xy-plane but also along the z-axis. To verify the output capability of the device along the vertical direction, z-axis excitation testing was conducted. Supplementary Fig. 42e shows the vertical testing platform, with the charge, current, and voltage for each electrode pair under optimal excitation conditions displayed in Supplementary Fig. 42f–i. It can be observed that

RD-TENG still demonstrates good output in the vertical direction, although slightly lower than that in horizontal motion. This is because when the device is placed vertically, the vibrating sheet's oscillator naturally sags under the influence of gravity, resulting in downward displacement and deflection before motion begins. During excitation, the oscillator is more likely to contact the lower fixed plate, whereas contacting the upper fixed plate requires overcoming gravity. As a result, in vertical testing, the efficiency of contact between the oscillator and the lower fixed plate is higher, while the contact efficiency with the upper plate is lower, leading to output attenuation. The azimuthal tests provide a basis for energy harvesting in the full space of RD-TENG with three linear degrees of freedom.

Under a 20 MΩ load and optimal combination, the 1–3 electrode pair achieved a peak power of 49.46 mW, with a VPD of 136.74 W m⁻³ (Fig. 3f). The matching impedance and peak power densities of other electrode pairs are provided in Supplementary Fig. 42j–l. Under 10 MΩ, 10 MΩ, and 20 MΩ loads, the peak powers reached 37.08 mW, 30.74 mW, and 41.24 mW, with VPD of 102.51 W m⁻³, 84.97 W m⁻³, and 114.00 W m⁻³, respectively. The 1–3 electrode pair exhibits superior output performance, attributable to its high-voltage output characteristics. Although the output of the other electrode pairs is slightly lower, it remains significantly higher than that of previous similar works (Table 1), thereby demonstrating the high efficiency of each electrode pair in mechanical energy harvesting when functioning as an output terminal, with the potential to power different electronic devices.

### Application in vibration energy harvesting

To further investigate the power supply capability of the RD-TENG for electronic devices under mechanical vibration energy harvesting, we charged capacitors and used them to power devices such as calculators, bicycle light warning, multimeters, and high-power commercial LED bulbs, under optimal conditions of 2.4 Hz frequency and 70 mm amplitude. The corresponding port combinations for the application-specific circuits discussed below can be referenced and configured based on Supplementary Fig. 43. Figure 3g presents the charging curves of 100 μF and 330 μF capacitors charged by the RD-TENG, with each electrode pair after rectification. The curve for the 100 μF capacitor shows that the power supply capability of each electrode pair is roughly the same. This is attributed to the relationship between voltage and current for each electrode pair: pairs with lower charge output exhibit higher voltage, while those with higher charge output show lower voltage. As the capacitance increased to 330 μF, the charging differences became more pronounced, with electrode pairs that produce higher charge output and current demonstrating a distinct advantage. To demonstrate the charging capability of the RD-TENG device with varying numbers of array elements, the left section of Fig. 3h shows the time required for 1–4 device arrays with rectifier electrode pairs to charge a 1 mF capacitor to 2.5 V. A single device required 226 s, while the four-device array achieved the same charge in just 91 s. The right section of Fig. 3h shows the charging curves of different capacitors charged by a single device rectifier electrode pair within 60 s. Within this time, a 47 μF capacitor was charged to 9.7 V, and a 220 μF capacitor to 2.3 V, demonstrating its good power supply capability. Supplementary Fig. 44a and Movie 2 demonstrate the ability of a single device to charge a 100 μF capacitor for 100 s and subsequently drive two parallel calculators. Figure 3i and Supplementary Movie 3 show that a 4-device array charged a 6.8 mF capacitor for 56 min to 9.7 V and subsequently powered two parallel multimeters rated for 9 V, with one multimeter used to measure the voltage of a 1.5 V dry cell battery. Supplementary Fig. 44b and Movie 4 show 1 to 4 device arrays with rectifier electrode pairs illuminating 32 2 W LED bulbs. The device arrays provide modular power supply capabilities, with light intensity increasing from dim to bright as the number of devices increases, reflecting a gradual enhancement in power supply

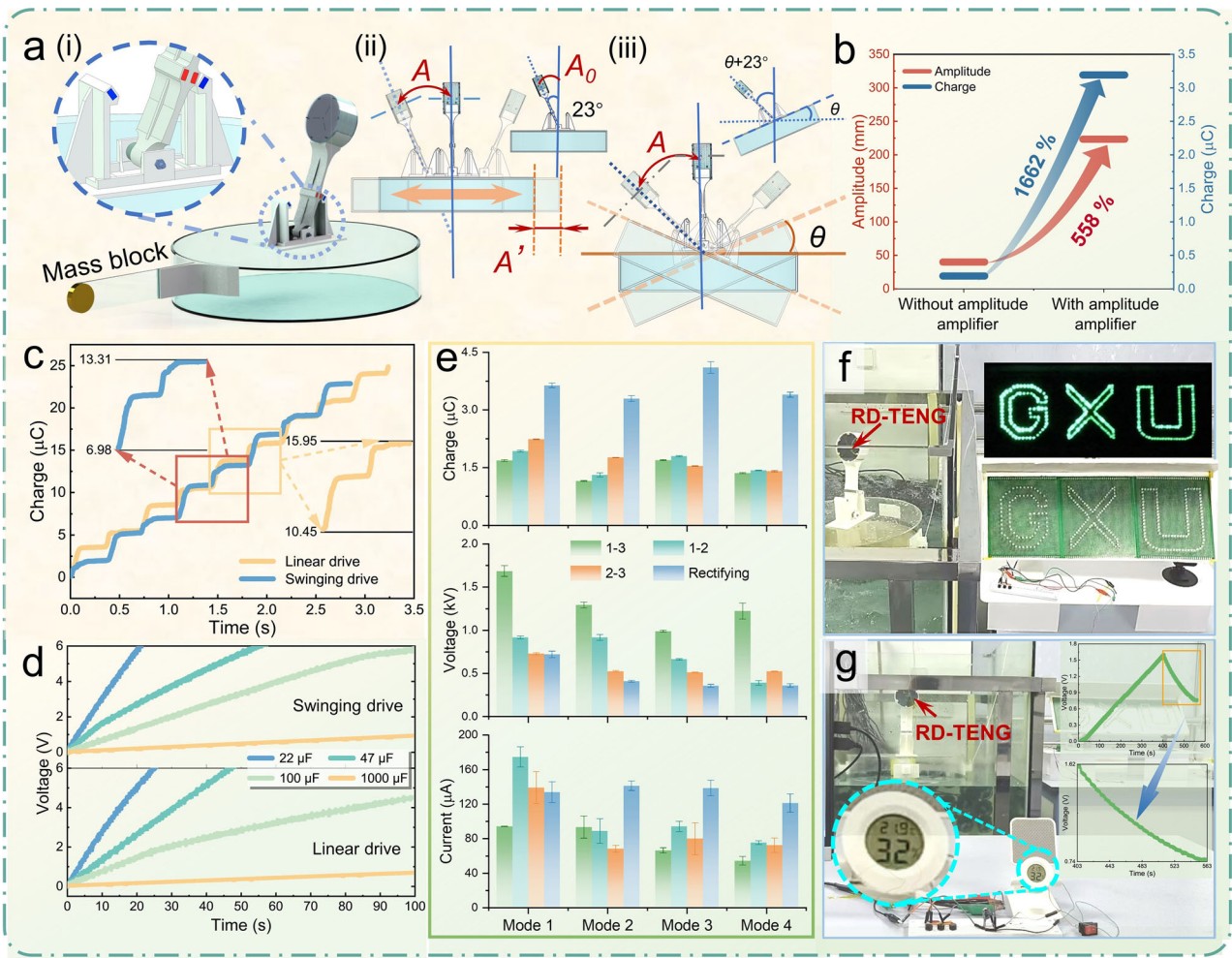

**Fig. 4 | Scenario adaptability expansion of the RD-TENG and wave energy harvesting study. a** Scenario adaptability expansion system, here, *A′* is the driven amplitude of the base, $A_O$ is the amplitude of the MRP, *A* is the amplitude of the RD-TENG, and *θ* is the phase angle of the driven base: (i) frequency-reducing and amplitude-amplifying MRP placed on the buoyancy base, (ii) linear drive mode, (iii) swinging drive mode. **b** Comparison of RD-TENG amplitude and output with and without the MRP under the same external excitation conditions. **c** Charge output of rectified electrode pairs under linear and swinging drive modes. **d** Charging curves of rectified electrode pairs with different capacitors under linear and swinging drive modes. **e** Charge, current, and voltage outputs of each electrode pair of the RD-TENG system under different driving intensity modes in the wave pool. Data are presented as mean values ± SD (*n* = 5). **f** Rectified electrode pair of the RD-TENG system is lighting up the "GXU" sign composed of 284 LEDs connected in series. **g** Rectified electrode pairs of the RD-TENG system charge a 2.2 mF capacitor to drive the temperature and humidity controller. Source data are provided as a Source Data file.

capability. Supplementary Fig. 44c also demonstrates the ability to drive the same row of bulbs under vertical testing conditions. Supplementary Fig. 44d and Movie 5 show the rectifier electrode pair of a single device driving five 2 W LED bulbs as a bicycle front light or warning light, providing safety alerts for activities such as night cycling or running. These applications demonstrate that the RD-TENG can effectively harvest mechanical vibration energy and power distributed electronic devices across multiple scenarios.

## Scenario adaptability expansion

Mechanical energy in nature often exists in low-frequency, low-amplitude, and irregular forms, such as ocean wave energy. To enable the RD-TENG to adapt to these environments and provide efficient energy harvesting, designing a system that can harvest energy at low frequencies and amplitudes is crucial. The RD-TENG requires higher frequencies and larger amplitudes under direct vibration excitation to achieve a good output. Therefore, we have designed a frequency-reducing and amplitude-amplifying magnetic repulsive pendulum (MRP) aimed at lowering the response frequency and amplifying the operating amplitude of the RD-TENG, thereby enhancing its ability to

capture ocean wave energy. As shown in Fig. 4a(i), the MRP is mounted on a cylindrical acrylic buoyant base and consists mainly of a pendulum rod and a support base (Supplementary Fig. 45). A two-square channel is embedded in the middle of the pendulum rod to house two powerful magnets, and a bearing is installed at the bottom to allow the pendulum rod to swing around its axis with a swing angle of 23°. The side deflector of the support base is provided with channels to place magnets with opposite polarity to the pendulum rod's magnet, generating a repulsive force that enables the pendulum rod to swing back and forth under external excitation. This repulsive force functions similarly to the restoring force of a spring. However, magnets are favored due to their superior response sensitivity, tunable dynamic characteristics, and mechanical and deployment advantages over traditional springs (Chapter 1 of Supplementary Note. 8). The support rod has two vertical surfaces and a small central hole, which is used to firmly connect screws, nuts, and bearings to the pendulum rod.

The interaction of ocean waves with objects can be summarized as a combination of linear motion along the XYZ axes and rotational motion around the XYZ axes in space. To investigate the output characteristics of the RD-TENG based on the MRP under these two

motion modes, we conducted the study using a six-degree-of-freedom (6-DOF) platform, which includes both linear and swinging drives. As shown in Fig. 4a(ii) and Supplementary Fig. 46a, the schematic diagram illustrates the RD-TENG system working along the horizontal direction under the 6-DOF linear drive. The movement of the base in the horizontal direction drives the MRP to swing, with the actual amplitude of the RD-TENG being $A = A_0 + A' > A'$ (where $A'$ is the external excitation amplitude and $A_0$ is the inherent swinging amplitude of the MRP), thus achieving amplitude amplification. As shown in Fig. 4a(iii) and Supplementary Fig. 46b, the schematic diagram illustrates the system swinging around its axis under the swinging drive. When the external excitation rotates the base by an angle $\theta$, the actual swinging angle of the device becomes $\theta + 23° > \theta$, thereby amplifying the swinging angle and further increasing the amplitude. The amplitude amplification calculations for both linear and swinging drive motion are detailed in Supplementary Fig. 47a, b. To visually demonstrate the amplitude amplification effect, we tested the charge output of the 1–3 electrode pair under 1.4 Hz, 20 mm linear drive amplitude, both with and without the MRP. Figure 4b shows that, compared to the case without the pendulum, the amplitude of the RD-TENG with MRP increased by 558%, with the charge output increased by 1662%, fully proving that the MRP significantly enhanced the output capability of the device by amplifying the actual motion amplitude. To further explore the output performance of the frequency-reducing and amplitude-amplifying system under these two drive modes, we conducted two additional orthogonal experiments (Supplementary Note. 7, Experiment 2). The results showed that, under the linear drive mode, the optimal output occurs at the best combination of 1.4 Hz frequency, 20 mm amplitude, and an amplitude factor of 4, with the influence order being frequency > amplitude > amplitude coefficient. The optimal combination for a swinging drive is: 1.4 Hz frequency, 10° angle, and an amplitude factor of 4, with the influence order being angle > frequency > amplitude coefficient. Based on this, we tested the output performance of each AC electrode pair under these two optimal combinations on the 6-DOF platform (linear drive shown in Supplementary Fig. 48a–c, swinging drive shown in Supplementary Fig. 48d–f). Under linear drive mode, the 1–3, 1–2, and 2–3 electrode pairs charge output 2.20 μC, 2.48 μC, and 3.19 μC, respectively. Under swinging drive mode, the outputs are 2.91 μC, 3.12 μC, and 3.29 μC, with excellent voltage and current performance in both modes. In Fig. 4c, the maximum charge accumulation per cycle for the rectifying electrode pair under linear drive mode is 5.48 μC, while under swinging mode, it reaches 6.33 μC. (Based on the three-electrode rectifying circuit connection method, the transferred charge per cycle under swinging mode is normalized to $6.33 \times 2/3 = 4.22$ μC, with a VCD of 11.69 mC m$^{-3}$ and a SCD of 42.23 μC m$^{-2}$). In summary, the swinging drive output is superior to the linear drive, attributed to the larger tilt angle under the swinging drive, allowing the vibrating sheet to unfold better under gravity. When in contact with the fixed sheet, a greater force results in more complete contact, leading to higher output (Supplementary Fig. 49). Supplementary Fig. 50 shows the comparison of the 1–3 electrode pair and the rectifying electrode pair lighting up 64 2 W LED bulbs under both linear and swinging drive modes, validating the power supply capability of different electrode pairs under both modes.

The maximum charge accumulation output increases from 5.5 μC with the linear motor to 6.33 μC with the 6-DOF swinging drive, while the frequency decreases from 2.4 Hz to 1.4 Hz, demonstrating that the MRP enhances the device's output performance and reduces the optimal operating frequency. To further showcase its frequency-reducing and amplitude-amplifying characteristics, we tested the variation of output with amplitude and frequency under both 6-DOF linear and swinging drive modes. As shown in Supplementary Fig. 48g, h, j, k, the output of both modes is largely unaffected by changes in external excitation parameters; once the excitation reaches the response threshold, it triggers the system's swing to

generate electrical energy, maintaining substantial output even at low frequencies down to 0.4 Hz, low amplitudes as small as 6 mm, and low response angles down to 4°. Without the MRP (on the linear motor), the 1–3 electrode pair outputs only 0.42 μC at 1.4 Hz and 70 mm, while the output of the 6-DOF system at 1.4 Hz and 6 mm linear drive reaches 2.2 μC. In the swinging mode at 0.4 Hz and 10° swinging angle, the output reaches 2.11 μC, demonstrating significant frequency-reducing and amplitude-amplifying characteristics (Supplementary Fig. 48i, 48l). To verify the durability of the RD-TENG under the MRP system, we conducted a long-term test under linear drive conditions. The 1–3 electrode pair was operated at 1.4 Hz with a 10 mm amplitude for 12,760 continuous cycles, exhibiting almost no performance degradation (Supplementary Fig. 51). This stability is attributed to systematic optimizations in mechanism design, structural configuration, and encapsulation strategy. Specifically, the introduction of the three-electrode charge dispatch mechanism effectively mitigates electrostatic shielding and charge cancellation, thereby enhancing charge transfer continuity and output uniformity. The device further benefits from the intrinsically low-wear nature of the contact-separation mode, ensuring long-term mechanical reliability. Moreover, the MRP provides non-contact and stable excitation, circumventing the fatigue issues typically associated with traditional elastic elements. Coupled with a high-density packaging design, the RD-TENG is able to operate reliably within a structurally stable and well-sealed environment. Figure 4d shows the charging curves for different capacitors by the rectifying electrode pair under linear drive (1.4 Hz frequency, 20 mm amplitude) and swinging drive (1.4 Hz frequency, 10° angle). The swinging mode demonstrates faster charging efficiency. Within 75 s, the swinging mode can charge a 100 μF capacitor to 5 V. Subsequently, we tested the wave energy harvesting capability of the MRP system in a laboratory wave pool. To enhance the wave-driven floating base effect, we added a mass block as an unbalanced gravity pendulum at the base's edge (Supplementary Fig. 52). As shown in the test results in Fig. 4e, the favorable output is observed under several excitation intensity modes in the wave pool (The parameters of each mode are shown in Supplementary Fig. 53), with Mode 1 yielding slightly higher output due to the excitation parameters are at their maximum, while the other modes show consistent and lower output, which is in line with previous analyses. This validates the system's outstanding performance in laboratory wave energy harvesting. Under Mode 1 drive, the rectifying electrode pair illuminated the "GXU" sign light group made of 284 series-connected LED lights, exhibiting significant light intensity (Fig. 4f and Supplementary Movie 6). Then, a 2.2 mF capacitor was charged to 1.6 V over 400 s, continuously driving a temperature and humidity meter for 156 s (Fig. 4g and Supplementary Movie 7), fully demonstrating the potential of the RD-TENG based on the unbalanced floating base and MRP in wave energy harvesting to power offshore devices.

## Field testing in oceanic conditions
In real marine environments, waves exhibit randomness and irregularity, with fragmented waves in various directions, posing challenges to the TENG-based wave energy harvesting system. The conversion of wave energy into electrical energy by TENG involves several stages: (1) energy absorption: transfer of wave energy to the device casing or wave-absorbing system, (2) energy transfer: from the wave-absorbing system to the TENG device, (3) energy conversion: the TENG device's output, and (4) Energy management and supply: output to energy storage or applications (Fig. 5a). Any failure in these stages would result in energy loss, significantly reducing the conversion efficiency of the TENG system. The RD-TENG system, based on an unbalanced buoyant base and the MRP, ensures efficient operation across all stages of marine testing, thereby safeguarding overall wave-energy harvesting performance: (1) an unbalanced buoyant base captures wave

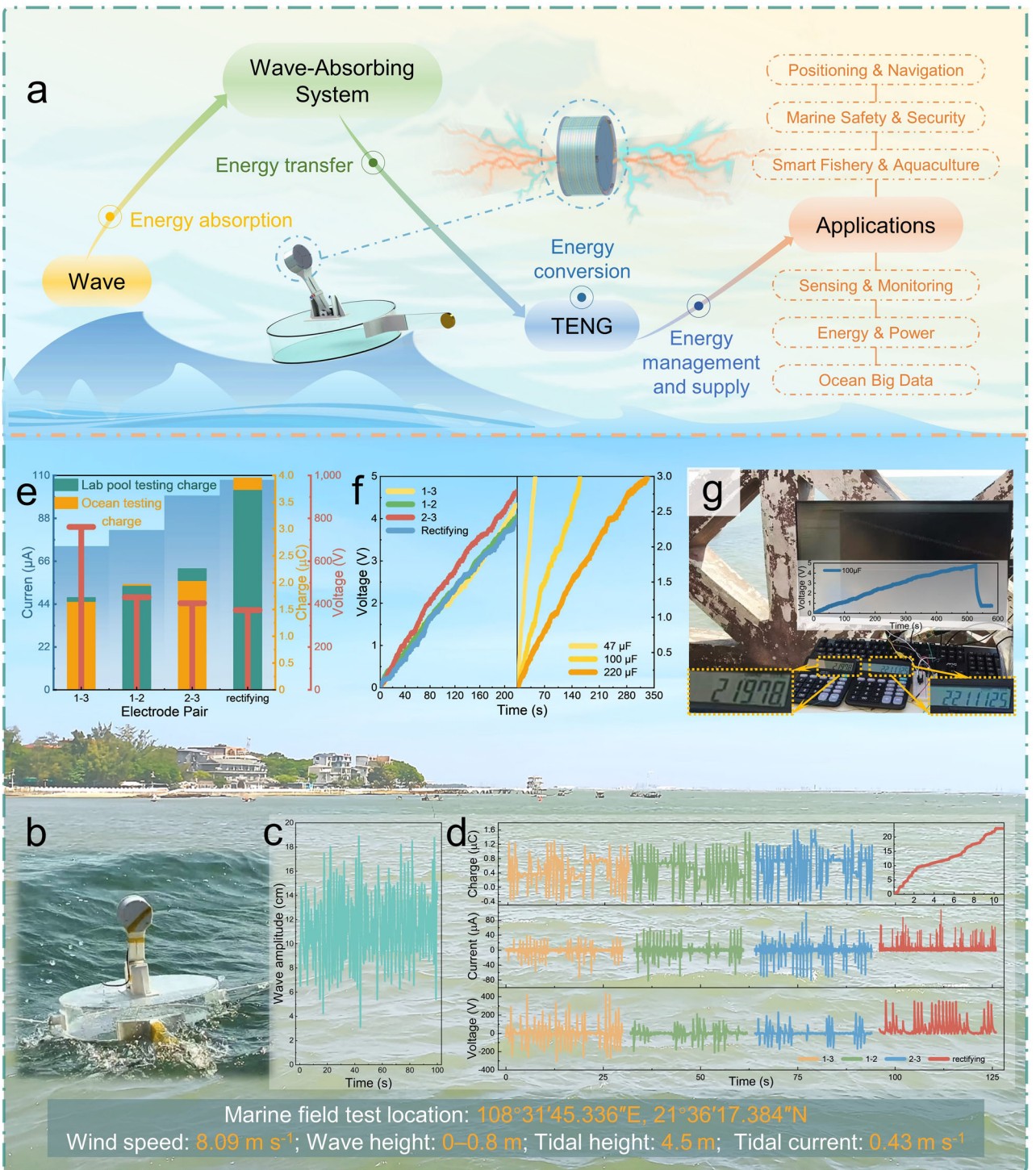

**Fig. 5 | Real-ocean wave energy collection using the RD-TENG system based on an imbalanced buoyancy base and an MRP. a** Technology roadmap of TENGs from water wave to marine IoT device applications. **b** Wave absorption motion of the RD-TENG system on the sea surface. **c** Sea wave amplitude data during testing. **d** Real output waveforms of charge, current, and voltage for each electrode pair. **e** Statistical data of charge, current, and voltage from the sea test, compared with the charge output in the wave pool under Mode 1. **f** Left: charging curves for each electrode pair with a 100 μF capacitor; right: charging curves for the 1–3 electrode pair with different capacitors. **g** After low tide, the system collects low-amplitude wave energy to power two calculators. Source data are provided as a Source Data file.

energy; (2) the MRP transfers the captured wave energy; (3) the RD-TENG device converts mechanical energy into electricity; and (4) the harvested energy is rectified, stored, and finally supplied to electronic devices. As shown in Fig. 5b and Supplementary Movie 8, we conducted field tests along the coastline of the Beibu Gulf, China (near 108° 31′ 45.336″ E, 21° 36′ 17.384″ N), showing the RD-TENG system's

wave absorption motion on the sea surface. Supplementary Fig. 54 shows real-time sea-state snapshots taken during data collection, illustrating mild-to-moderate sea conditions. Under these conditions, the RD-TENG and its buoyant platform maintained stability without short circuits or structural damage, as demonstrated in Supplementary Fig. 55 (Potential limitations and possible mitigation strategies under

extreme sea conditions are discussed in Chapter 3 of Supplementary Note. 8). Furthermore, the system exhibited wave-following response capability. Figure 5c shows the real measurement wave amplitude data near the base on that day, with a peak amplitude of 16 cm, though most of the time, the wave amplitude was between 7 cm and 10 cm. Figure 5d displays the actual data waveform of the system generating power by absorbing wave energy at these wave amplitudes. The data shows that the charge, current, and voltage for the 1–3, 1–2, 2–3, and rectifier electrode pairs are as follows: 1.65 μC, 73.72 μA, 760 V; 1.97 μC, 82 μA, 432 V; 2.04 μC, 99.75 μA, 404 V; 3.95 μC, 107.75 μA, 372 V. The output relationships between the electrode pairs are consistent with the conclusions derived from laboratory tests. Figure 5e shows the statistical data of charge, current, and voltage from the marine test, comparing them with the laboratory charge output data (Mode 1). The charge output from the marine test is comparable to the best laboratory output. To demonstrate the system's power supply capability, we tested the charging of the same capacitor by different electrode pairs and the charging of different capacitors by a single electrode pair. As shown in the left part of Fig. 5f, the four electrode pairs charged a 100 μF capacitor within 220 s, with the 2–3 electrode pair reaching 4.7 V slightly faster, while the charging capabilities of the other three electrode pairs were similar. The right part shows the time required for the 1–3 electrode pair to charge different capacitors to 3 V. It is evident that a 220 μF capacitor can be charged to 3 V within 328 s, highlighting the system's real-world power supply capability. To evaluate the RD-TENG system's energy conversion efficiency under real-sea conditions, we compared capacitor charging performance (100 μF, rectified electrode pair) against optimal laboratory conditions on the 6-DOF platform. After 1 min of charging, the voltage reached 1.09 V at sea vs and 4.06 V in the lab, achieving approximately 26.85% of laboratory efficiency (Supplementary Fig. 56). This result indicates substantial headroom (up to 71.75%) for performance enhancement under stronger marine excitation. However, due to the inherent randomness and irregularity of natural wave conditions, TENG triggering frequency and operation efficiency at sea were inevitably lower than under controlled lab settings (Supplementary Fig. 57). To validate the system's ability to power electronic devices under low-frequency and low-amplitude conditions, Fig. 5g and Supplementary Movie 9 demonstrate that after charging a 100 μF capacitor to 4.8 V, the system smoothly powered two calculators for 20 s during low tide with low waves. This fully validates the RD-TENG system, based on the unbalanced buoyant base and MRP, for its efficient wave-to-electric energy conversion and power supply capabilities for marine electronic devices under low-frequency and low-amplitude conditions, further proving its energy harvesting adaptability across various scenarios, and expecting to promote the development of the marine IoTs. In the next phase, we aim to further integrate the RD-TENG into critical marine applications such as wireless communications, remote positioning, and navigation guidance, establishing a more rigorous validation workflow. Ultimately, we strive to drive the practical and scalable deployment of a self-powered blue energy IoT network.

## Discussion

This study presents a rattle drum-inspired TENG (RD-TENG) based on the charge dispatch strategy, for which a comprehensive theoretical quantitative model is further established. This strategy constructs internal and external charge circulation paths via a dual-TENG configuration, enabling charge diversion to overcome cancellation issues arising from shared electrodes in dense FCS stacked models, and converts the fundamental unit from a single-electrode mode to a contact-separation mode to mitigate the inherent electrostatic shielding. This results in a 6-fold increase in output compared to conventional models, achieving a volumetric peak power density of 136.74 W m$^{-3}$ and successfully powering various electronic devices. Structural innovations address contact-separation inefficiencies,

including laser-etched integrated vibrating steel sheets, contact push-pin conduction, and a looping stacking method, enhancing layer density, space utilization, cost-effectiveness, and output performance, reaching a TSD of 2.76 cm$^{-1}$. The FCS mode and spring steel ensure mechanical durability, while a rotating-clasp parallel array design increases energy harvesting density and stability for large-scale applications. Additionally, the RD-TENG incorporates a frequency-reducing and amplitude-amplifying MRP, resulting in motion amplitude and output charge increases of 558% and 1662%, respectively. This reduces the wave energy collection frequency band to 0.4 Hz, the amplitude response to 6 mm, and the angle response to 4°. Field tests at China's Beibu Gulf validated its performance, with short-circuit charges of 1.65 μC, 1.97 μC, and 2.04 μC per AC electrode pair. Successfully powered electronic devices under mild sea conditions. In conclusion, this work introduces a charge dispatch strategy and scalable structural designs, enhancing TENG output performance and scenario adaptability, providing a foundation for the industrial and commercial development of TENG technology.

## Methods

### Fabrication of the RD-TENG device

The RD-TENG device consists of a core power generation component and a packaging shell. The core includes 8 fixing sheets, 7 vibrating sheets, 14 acrylic ring gaskets, 1 PET insulating ring, 3 push pins, 3 latches, 3 electrode sheets, and 3 conductive sponges. The packaging shell is made up of a bottom plate, a packaging barrel, and a top plate, with a rotating-clasp device required for array integration.

The electrode of fixing sheets, vibrating sheets, and electrode sheets are laser-etched from Mn 65 silicon-manganese spring steel (the rationale for material selection is provided in Chapter 1 of Supplementary Note. 8, while the existing anti-corrosion techniques and prospective optimization strategies are discussed in Chapter 3 of Supplementary Note. 8), each with a 96 mm outer diameter and six alternating holes (three for electrode connections and three for latch fixation). A 0.08 mm PTFE film is adhered to one side of each fixing sheet, while vibrating sheets come in two types: 4 fully coated and 3 uncoated PTFE film sheets. Each vibrating sheet features a fixed outer ring, inner oscillator, and three cantilever bridges. Acrylic ring gaskets (2.75 mm thick) space the oscillator's movement, while a 0.5 mm PET insulating ring ensures insulation. Push pins (1 mm diameter) are made of carbon steel for high conductivity, while latches, laser-cut from acrylic, secure the layers. Electrode sheets (0.1 mm thick) are placed on the insulating ring to align with the push pins, and a 2 mm thick conductive sponge is fitted into the top plate's reserved holes. The packaging barrel is 3D-printed from PLA. Assembly proceeds as follows: The bottom plate is prepared, followed by the insertion of electrode push pins and latches. The right PTFE-coated fixing sheet (electrode 1) is placed, followed by a ring gasket, an uncoated vibrating sheet (electrode 2), another gasket, and the left PTFE-coated fixing sheet (electrode 3). The process is repeated in sequence until the assembly is complete, then the insulating ring, electrode sheet, packaging barrel, top plate, and conductive sponge are added. The top and bottom plates are sealed to the packaging barrel with waterproof adhesive to complete the RD-TENG.

For array integration, a 3D-printed rotating-clasp device (100 mm inner diameter) is used. It features locking slots to align with the packaging barrel's protrusions and a central viewing window for connection monitoring. The clasp is fastened to one RD-TENG, and another is aligned with the conductive sponge before being inserted and rotated to form the integrated array.

### Fabrication of the frequency-reducing and amplitude-amplifying MRP

The frequency-reducing and amplitude-amplifying MRP consists of two parts, both fabricated using 3D printing technology. The first part

is the pendulum rod, and the second part is the support base (Supplementary Fig. 45). The upper part of the pendulum rod is used to fix the RD-TENG device, which can be directly bonded to the rod using adhesive. Square channels are provided on both sides of the middle section of the pendulum rod, each containing a magnet (N52 NdFeB magnets). The circular channel at the bottom is used to accommodate the bearing, which connects to the support base, allowing the pendulum rod to swing linearly around the center of the bearing. The support base is equipped with baffles on both the left and right sides. Channels at the top of the baffles are used to place magnets, so that when the pendulum rod swings to either side, the magnet in the middle of the rod repels the magnets at the top of the baffles, causing it to bounce back and forth, thus realizing a repetitive motion. The center of the support base has two vertically arranged circular holes, through which the pendulum rod is connected to the support base using a screw (M4 × 45 mm), a nut (inner diameter 4 mm, thickness 3 mm), and a bearing (outer diameter 16 mm, inner diameter 4 mm, thickness 5 mm). The other beam and column structures of the pendulum rod and support base are designed based on engineering mechanics and cost reduction considerations.

## Manufacture of the unbalanced buoyancy base

The main body is made of an acrylic cylindrical barrel with a diameter of 500 mm and a height of 100 mm, with a wall thickness of 5 mm. Vertical cantilever beams, 3D printed, are fixed to the outer edge of the barrel. At the end of the cantilever beams, stainless steel counterweights are placed. Under the leverage effect of the counterweight, the buoyancy base is in a more unbalanced state, allowing the MRP on top to swing more effectively, thereby exerting a higher and more efficient output on the TENG.

## Measurement methods and instruments

The P01-37 × 120-C/C1100 linear motor is used to simulate motion energy/vibrational mechanical energy, while the Keithley 6514 system ammeter is used to measure the transferred charge and short-circuit current of the TENG. An oscilloscope (Tektronix MDO3012) and a high-voltage probe (Tektronix P6051A) are used to measure the open-circuit voltage. A six-degree-of-freedom motion platform (WIN-020, 33 KW) is employed to provide uniform and adjustable linear and swinging drive. For the water-based laboratory tests, a wave pool with dimensions of 1.2 m × 1 m × 1 m is installed with a 3 × 10 array of 30 wave-making pumps (JieBao, RW50, 50W) arranged in a vertical plane.

## Data availability

All data needed to evaluate the conclusions in the paper are presented in the paper. Any additional requests for information can be directed to and will be fulfilled by the corresponding authors. Source data are provided with this paper.

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

## Acknowledgements

The research was supported by the National Natural Science Foundation of China (62304058 and 52362025), the National Key R&D Project from the Ministry of Science and Technology (2021YFA1201603), and the Innovation Project of Guangxi Graduate Education (YCBZ2023038).

## Author contributions

W.T. and G.L. conceived the idea, designed and fabricated the device, and wrote the manuscript. W.T. performed experiments and analyzed the data. H.L., W.Z., J.D. and Y.W. helped with the experiments. J.L. helped conduct a potential field simulation and construct a quantitative theoretical model. G.L. reviewed and corrected the manuscript. G.L. and L.W. supervised and guided this work. All authors discussed the results and commented on the manuscript.

## Competing interests

The authors declare no competing interests.
