## [Transparent Peer Review file · Nature Communications]

Rattle Drum Inspired Triboelectric Nanogenerator with Enhanced Output Using Charge Dispatch and Magnetic Repulsion Pendulum

Corresponding Author: Dr Guanlin Liu

Version 0:

Reviewer comments:

Reviewer #1

(Remarks to the Author)

In the manuscript, the authors present a meticulously engineered Rattle Drum inspired Triboelectric Nanogenerator (RD-TENG) with a notably high Triboelectric Surface Density (TSD). The manuscript and its extensive Supplementary Information detail innovative structural designs (e.g., laser-etched vibrating sheets, push-pin connections, layer-stacking methods), a systematic optimization approach using orthogonal experimental design. While the work demonstrates engineering advancement, several fundamental concerns limit its suitability for prestigious journal as Nature Communications. Detailed comments are as below:

- The claimed "novel charge dispatch strategy" seems to represent an incremental advancement rather than a breakthrough innovation. The work primarily represents engineering optimization rather than fundamental scientific advancement. While achieving higher outputs through structural modifications has practical value, it does not advance our understanding of triboelectric phenomena, contact electrification mechanisms, or energy conversion principles. The manuscript reads more like a device optimization study than a contribution to fundamental science. The manuscript explicitly states, "The integration of three-electrode technology and alternating film coatings allows free electrons to shuttle between the internal electrodes...". Three-electrode TENG designs have been previously reported, and the fundamental working principle—using internal charge redistribution to enhance output—lacks conceptual originality. A breakthrough innovation would typically introduce a fundamentally new physical principle or TENG operational mode.
- The innovation lies in the specific structural design (rattle-drum mechanics, alternating coatings, push-pin connections) and the particular way the three electrodes are configured and interact—especially the role of the short-circuited internal electrode 2 as a "charge dispatch channel". This is a refinement and specific application of existing concepts, characteristic of an incremental advancement. The authors themselves frame their work as combining "alternate film coating and a three-electrode design" as well. The "rattle drum inspiration" appears more metaphorical than mechanistically meaningful, serving primarily as branding rather than providing genuine biomimetic insights.
- The emphasis on triboelectric surface density (TSD) as a primary performance indicator is scientifically questionable. Defining TSD as surface area per unit volume creates a metric easily manipulated through geometric optimization rather than fundamental material or physical improvements. This appears to be metric engineering rather than genuine performance enhancement. The claimed "record" TSD of 2.76 cm^{-1} may simply reflect device miniaturization rather than superior energy conversion efficiency.
- The comparisons with "traditional models" lack fairness. The paper does not adequately demonstrate that compared devices represent state-of-the-art optimized designs. Previous literature shows charge densities reaching $1250 \mu\text{C m}^{-2}$ (<https://doi.org/10.1038/s41467-022-33766-z>) in controlled conditions and 5.4 mC m^{-2} for DC-TENGs (<https://doi.org/10.1038/s41467-020-20045-y>), suggesting the performance improvements claimed here are not as exceptional as presented. Without a clear presentation and comparison of the RD-TENG's intrinsic surface charge density, it is difficult to ascertain whether the reported high volumetric outputs stem from a fundamental breakthrough in charge generation/separation or primarily from the device's successful geometric densification. The authors should provide this data and discussion.
- The COMSOL simulations of electric potential provide a qualitative visualization of the proposed charge dispatch mechanism. However, for a significant advance, a more comprehensive theoretical framework is expected. This would ideally include rigorous electrostatic analysis that quantitatively predicts the reduction in electrostatic shielding and the enhancement in potential difference due to the specific three-electrode configuration and charge dispatch strategy, moving

beyond phenomenological descriptions.

- The extensive demonstrations of wave energy harvesting, including under real-ocean conditions (Fig. 5), are impressive and showcase the practical potential of the engineered RD-TENG system. While this applied aspect is a strength, its contribution to the manuscript's suitability for the journal will heavily depend on the perceived significance and fundamental novelty of the core TENG device itself. If the core TENG mechanism is considered a sufficiently innovative advance, the application robustly underscores its impact.

Reviewer #2

(Remarks to the Author)

This work presents a novel triboelectric nanogenerator inspired by the rattle drum (RD-TENG) that can overcome the limitations of output performance due to electrostatic shielding and low displacement amplitude in high density TENGs. In addition, it can enhance the output by raising the triboelectric surface density (TSD). In this design, a three-electrode configuration combined with alternating film coatings, enabling charge dispatch and electrostatic potential balancing is proposed. For better contact-separation, traditional springs is replaced by a laser-etched elastic steel sheet. Also, a push-pin contact system and Lego-like modular locking for easy assembly into arrays are also introduced.

Some key achievements are highlighted such as the record TSD of 2.76 cm^{-1} , the output volage is up to 2200 V, and peak power density is of 136.74 W.m^{-3} , it is 6 -fold output versus traditional models.

The manuscript can be accepted after addressing the following comments:

Major comments:

1. Is it possible to use the three-electrode charge dispatch strategy for other TENG system design?
2. Why were spring steel chosen as the vibrating element, given that it is susceptible to corrosion over time in marine environments?
3. Please prove the stable charge transfer and the performance of RD-TENG after long-term operation?
4. Please explain the optimal condition at 2.4 Hz and 70 mm amplitude, how if the higher frequencies or amplitudes provide higher output performance?
5. It looks like the RD-TENG operated under the nice ocean weather conditions, how about its efficiency under harsh ocean environments? Such as strong turbulence condition.
6. In this manuscript, the authors mention the technology roadmap of TENGs from water wave to marine IoT device applications, however, they are not demonstrated in the application part. Which specific IoT or sensing system you are targeting to show with this design?
7. It is important to maintain synchronization and the stability of stacking multiple devices, how can you solve this issue?

Minor comments:

1. In figure 1a-b-c, it is a need to explain clearly about the sheet 1-2-3, might be state them as fixing electrode 1, vibrating electrode 2, and fixing electrode 3?
2. In Figure 1e, it is a schematic of RD-TENG including 7 free-standing layer vertical contact-separation mode TENG (FCS-TENG). I suggest simplifying the schematic into 2 units at both end of RD-TENG, the middle can keep blank with 3 horizontal dots to simplify the schematic, easier for the reader.
3. Is it duplicated figure between Figure 1 and Figure Note S2? The current value shown at the voltmeter are different, while the schematic showing the vibrating electrode 2 is at same position. Please clarify this issue.
4. All the arrows in figures are not readable, please change it to be clearer and simple to the reader. In addition, in Figure 2a, the RD-TENG can be shown in cross-sectional view for better presentation.

Reviewer #3

(Remarks to the Author)

The paper is very well written, The TENG design shows a fair amount of novelty and the results are interesting. Therefor, I believe that the paper could be worthy of publication after a few adjustments are suggested:

1. I believe that there are statements in the introduction lack references, for example:

Line 43, Page 2: "However, the path to the commercialization of TENGs is fraught with stumbling blocks, including suboptimal output performance, limited mechanical lifespan, high fabrication costs, and limited adaptability to various environments"

Line 48, Page 2: "This disagreement stems not only from the significant variations in practical applications but also from the lack of a universally accepted, straightforward metric for evaluating the performance of different TENG structures. The introduction of triboelectric surface density (TSD) has, to some degree, provided a resolution to these controversies."

2. In figure 2b, it would be good to label that the fixed and vibrating sheets are electrodes (similar to the labels in figure in Note S2).

3. In page 13, Line 347. I would like to see a better explanation for why repulsive magnets were used in the structure of the magnetic repulsive pendulum (which I assume is acting like a spring).

Version 1:

Reviewer comments:

Reviewer #1

(Remarks to the Author)

The authors have answered the comment from the last revision in detail. Responses from last revision provides a more robust and nuanced understanding of its contributions.

Reviewer #2

(Remarks to the Author)

Reviewer #3

(Remarks to the Author)

I would like to note that there are instances were the word "sheet" was misspelled as "sheep" (lines 120 and 135). Other than that, I believe that the revised version of this manuscript addressed the concerns raised in my previous revision, and I therefor can recommend this manuscript for publication.

REVIEWER COMMENTS

Response: We sincerely thank all reviewers for your time, expertise, and constructive insights in reviewing our manuscript. Your comments have provided invaluable guidance for deepening our scientific discussion, enhancing mechanistic clarity, also significantly contributed to the overall improvement of the manuscript.

To facilitate navigation, we have marked and tracked the revisions as follows: All cited references now include hyperlinks for direct access to source webpages. Reviewer comments and original manuscript content in black. Our point-by-point responses are in blue, and highlighted sections are in red. Modifications in the revised manuscript are in purple. In the revised version, we made the following major updates:

1. Identifies challenges in conventional models under high-density stacking, with analysis in Supplementary Note 1.
2. Explains the theoretical pathway from rattle-drum dynamics to RD-TENG electrodynamics in Notes 2 and 3.
3. Highlights originality; Note 4 compares the triple-electrode design with previous three-electrode models.
4. Establishes an equivalent capacitance quantitative theoretical model with analytical solutions; forms a validation loop with Fig. 1g–h and Note 5.
5. Introduces a comprehensive evaluation framework centered on TSD; adds Fig. 1i and Note 6 comparing multiple metrics.
6. Adds Note 8 covering materials, design strategy, scalability, potential risks, and future improvements.
7. Refines data analysis and adds literature support for clearer trend interpretation.
8. Improves figures and captions, adds sea-trial visuals and efficiency details, and refines technical aspects throughout.
9. The manuscript has been formatted following the journal's *Guide to Formatting Articles*.

In parallel, we revised the abstract, introduction, and results sections to refocus on the core physical mechanism. Once again, we sincerely thank the editor and all reviewers for your thoughtful comments, valuable suggestions, and dedicated efforts. We hope that the revised manuscript, with its strengthened theoretical framework, clarified contributions, and expanded validation, meets your expectations. We deeply appreciate your time and support, and we extend our best wishes to you all.

Reviewer #1 (Remarks to the Author):

In the manuscript, the authors present a meticulously engineered Rattle Drum inspired Triboelectric Nanogenerator (RD-TENG) with a notably high Triboelectric Surface Density (TSD). The manuscript and its extensive Supplementary Information detail innovative structural designs (e.g., laser-etched vibrating sheets, push-pin connections, layer-stacking methods), a systematic optimization approach using orthogonal experimental design. While the work demonstrates engineering advancement, several fundamental concerns limit its suitability for prestigious journal as Nature Communications. Detailed comments are as below:

Response: We sincerely thank the reviewer for the time and effort dedicated to the thorough evaluation of this work, and for accurately identifying the engineering innovations in our structural design—such as laser-etched vibrating sheets, push-pin connections, and layer-stacking methods. These designs not only endow the device with excellent output performance, scalability, and engineering value but also significantly enhance its potential for real-world deployment.

We acknowledge that the original manuscript underemphasized the core physical mechanism behind the structural innovations. The true breakthrough lies in the first implementation of a three-electrode charge dispatch strategy within a dense FCS-TENG. By embedding a short-circuited TENG as an internal electrode, this approach enables microscopic charge diversion and macroscopic mode transformation, fundamentally overcoming charge cancellation and electrostatic shielding effects inherent to traditional stacked TENGs. The manuscript now fully reflects both the fundamental mechanism innovation and the engineering viability of our strategy. We deeply appreciate the reviewer's insightful and impactful comments, which prompted us to systematically reflect on the scientific positioning of this work and present a more complete, mechanism-driven explanation of its theoretical significance and practical value within the TENG field. We hope this revised version more effectively conveys the depth and potential impact of our research and earns your continued recognition.

1. The claimed "novel charge dispatch strategy" seems to represent an incremental advancement rather than a breakthrough innovation. The work primarily represents engineering optimization rather than fundamental scientific advancement. While achieving higher outputs through structural modifications has practical value, it does not advance our understanding of triboelectric phenomena, contact electrification mechanisms, or energy conversion principles. The manuscript reads more like a device optimization study than a contribution to fundamental science. The manuscript explicitly states, "The integration of three-electrode technology and

alternating film coatings allows free electrons to shuttle between the internal electrodes...". Three-electrode TENG designs have been previously reported, and the fundamental working principle—using internal charge redistribution to enhance output—lacks conceptual originality. A breakthrough innovation would typically introduce a fundamentally new physical principle or TENG operational mode.

Response: We thank the reviewer for recognizing the engineering utility of our work. As noted, output enhancement alone is insufficient without fundamental mechanism innovation. This study proposes the first three-electrode charge dispatch strategy in dense FCS-TENGs, enabling charge diversion and mode transformation to solve long-standing shielding and cancellation issues. We clarify the mechanism, establish a quantitative model, and validate it through simulation and experiments. The design also supports scalability, multi-scenario adaptation, and real-sea deployment, offering a systematic contribution to the TENG field:

1. Development Trend of TENG Structural Devices

Densification and modular stacking have become key trends in TENG development (Adv. Mater. Technol. 2025, e00184), analogous to integration strategies in microelectronics and other energy systems. Future distributed energy networks demand compact, low-cost, and scalable TENG units.

2. Structural Bottlenecks under this Trend

As the structure becomes denser, a series of performance bottlenecks arise, including reduced separation displacement, charge cancellation due to shared electrodes, and electrostatic shielding inherent to single-electrode units (Supplementary Note 1). These issues parallel bottlenecks in Moore's Law scaling and electromagnetic systems, where high-density integration leads to performance loss unless systemic innovations are introduced.

3. Innovative Strategies to Overcome Output and Structural Bottlenecks

1) (Novelty #1 and plays a critical role) Resolving Charge Cancellation and Electrostatic Shielding via Charge Dispatch Mechanism

(The detailed theoretical quantitative model is elaborated in our response to your 5# comment)

i) Path Diversion to Overcome Charge Cancellation

As shown in **Fig. R#1-1**, FCS-mode TENGs are often stacked by isolating each unit with insulating layers to avoid electrode-sharing, aiming to reduce electrostatic shielding and charge cancellation (Supplementary Note 1). Prior works (e.g., Sci. Adv. 11, eadv9379 (2025), Nano Energy 2023, 116, 108818, Nat. Commun. 2025, 16 (1), 5141.) adopt this approach, but these isolated stacks lack internal coordination, cannot fundamentally resolve shared-electrode-induced charge losses, waste of space and low output.

Fig. R#1-1. Feasible configurations of traditional FCS-mode stacked devices.

Unlike previous FCS-stacked TENGs that avoid electrode sharing via insulating layers, we propose a three-electrode system featuring a vibrating short-circuit electrode (Electrode 2) that dynamically bridges internal and external loops. This enables charge path diversion rather than simple stacking, breaking shared-electrode-induced charge cancellation. While past works added extra electrodes for signal enhancement, our structure fundamentally redefines charge flow through coordinated internal–external circulation, enabled by alternate film coating and dual-path interfaces.

ii) Mode Transformation to Overcome Electrostatic Shielding

Beyond addressing charge cancellation via dual-path dispatch, RD-TENG mitigates electrostatic shielding through a fundamental unit-level mode transformation—from conventional single-electrode (SE) mode to contact-separation (CS) mode—by introducing a short-circuit electrode pair. This enables each layer to function as an independent CS unit, overcoming severe shielding in dense layer stacks.

As shown in Fig. R#1-2, while FCS mode have been widely used in single-layer TENGs (e.g., Nano Energy 2025, 133, 110437, Nano Energy 90 (2021) 106503), prior multilayer designs avoid shared electrodes to prevent path interference and shielding. To date, no report has demonstrated vertically stacked high-density FCS units with shared electrodes.

Here, by combining alternate film coating with a three-electrode charge dispatch strategy, we realize the first system-level multilayer FCS architecture featuring independently vibrating layers, shared electrodes, and decoupled output paths. This breakthrough enables scalable stacking of CS units, marking a key step toward high-density TENGs with practical deployment potential. Owing to its synergistic multilayer response and modular integration capability, this architecture shows great promise for scalable vibration energy harvesting and sensing, especially in complex environments such as industrial machinery, flow-induced systems, and infrastructure monitoring.

Fig. R#1-2. Schematic of a traditional single-unit, independently layered vertical contact-separation mode.

2) (Novelty #2) Addressing the Reduced Contact-Separation Efficiency in High-Density Stacked Structures through Structural and Engineering Innovations.

We developed a series of scalable and modular structural strategies to improve contact-separation efficiency and internal layout, simplify assembly, and support automation:

- i) Optimized Design of Vibrating Sheets;
- ii) Contact Push Pin–Based Electrode Connection;
- iii) Looping–Method Push-Fit Assembly of Triboelectric Units.

4. The charge dispatch mechanism is supported by a complete and mutually validated chain of theoretical evidence.

The proposed mechanism is underpinned by a complete theoretical framework with 4 layers of validation: 1) Analysis of charge transfer principles; 2) Finite element simulation; 3) Construction of a theoretical quantitative model (Newly added). 4) Model testing comparison.

5. Scalable Array Integration and Scenario Adaptability Expansion

1) (Novelty #3) Rotating-Clasp Array Integration Method. We developed a modular rotating-clasp design that enables fast, LEGO-like stacking of RD-TENG units into dense arrays. Each unit connects mechanically via a rotating clasp and electrically via a push pin–sponge–electrode sheet–pin chain, ensuring stable signal transmission across all three electrodes. This approach supports scalable deployment, modular maintenance, and robust system-level assembly.

2) (Novelty #4) Scenario Adaptability Expansion To address low-frequency excitation limitations, we introduced a magnetic repulsion pendulum platform that converts ocean-like wave motion into large-amplitude vibration, driving the RD-TENG efficiently. A full energy conversion chain from wave to device output was established and experimentally validated. This strategy broadens TENG applicability to real-world, irregular environments.

6. Advancement of a Comprehensive Evaluation Framework

(Details of the comprehensive evaluation framework in our response to your 3# comment)

7. Introduction of a New Experimental Method and Extensive Validation

We first introduced the orthogonal test method to optimize structural and excitation parameters

in TENGs' field, facilitating the exploration of mech-elect conversion in devices. RD-TENG's performance was verified on 4 platforms: 1) A linear motor setup powered high-load devices like 2 W LEDs, calculators, and multimeters. 2) A 6-DOF motion system and 3) a wave tank verified adaptability under complex wave-like excitations, enabling the power of various high-power electronic devices. 4) In real-sea tests in Beibu Gulf, RD-TENG showed stable operation and energy harvesting even under low-frequency, low-amplitude wave conditions.

We sincerely appreciate the reviewer's emphasis on mechanistic originality. As clarified, RD-TENG contributes not only through performance gains but by establishing a system-level mechanism for high-density TENGs—centered on mode transformation, charge path diversion, and structural innovation. Supported by modular and scenario-adaptive designs, this framework advances TENGs toward scalable platform integration. We will further highlight this mechanism-structure-system synergy and provide literature comparisons to better position our work within the TENG field.

Revision in *Main Manuscript*:

Abstract

Page 1, Line 7-18: ... However, structural densification imposes limitations that hinder further output enhancement. A charge dispatch strategy is developed in our proposed rattle drum inspired TENG (RD-TENG), which mitigates charge cancellation via path diversion and overcomes electrostatic shielding through mode transformation, yielding over 6x output versus traditional models. Further innovations to improve layers' contact-separation efficiency, including laser etching and the contact push pins method, raise the triboelectric surface density (TSD) to a record 2.76 cm^{-1} . A complete framework is established for RD-TENG, spanning theoretical modeling, structural design, and comprehensive experimental validation. Furthermore, ...

Introduction

Page 2, Line 47-75, Line 86-93: ... In addition, when combined with the Specific Surface Area (SSA),¹⁷ the triboelectric surface area per unit mass—it enables evaluation of the device's lightweight design. Together, TSD and SSA constitute key structural performance metrics for TENG devices. Improving TSD is expected to overcome current technical bottlenecks and accelerate TENG industrialization. Compactness, triboelectric layer integration, and lightweight design are regarded as future trends for distributed energy devices.¹⁷ However, as TSD and triboelectric layer density increase, achieving a sufficient contact-separation distance

becomes increasingly challenging. Furthermore, the traditional structurally decomposed fundamental unit is the single-electrode mode, which inherently suffers from electrostatic shielding effects;³¹ resulting in device internal electrostatic shielding intensifies with increasing layer density, and in free-standing vertical contact-separation (FCS) configurations, the use of shared electrode layers leads to severe charge cancellation between stacked units, further limiting the overall electrical output. Therefore, ...

In this work, we introduce a novel rattle drum inspired triboelectric nanogenerator (RD-TENG), ... , realizing microscale charge diversion and macroscale mode transformation of the basic working units. This design effectively mitigates the electrostatic shielding effects and charge cancellation effects, leading to a high output at the external electrode. Combining structural innovations based on this strategy to improve contact-separation efficiency, yielding a TSD of 2.76 cm^{-1} for the RD-TENG. ...

..., the five-in-one evaluation system, including structural and electrical performance indicators, comprehensively surpasses previous results (Fig. 1k-1l).³²⁻⁴⁶ ...

Discussion

Page 21, Line 534-540, 547, 550-551: This study ..., which constructs internal and external charge circulation paths via a dual-TENG configuration and alternate film coatings. This enables charge diversion to overcome cancellation issues arising from shared electrodes in conventional high-density stacked models, and converts the fundamental unit from a single-electrode mode to a contact-separation mode to mitigate the inherent electrostatic shielding in dense FCS devices. ... Structural innovations address contact-separation inefficiencies, including ... achieving a VCD of 11.69 mC m^{-3} and SCD of $42.23 \text{ } \mu\text{C m}^{-2}$. Field tests at China's Beibu Gulf ... Based on these results, we further established a comprehensive evaluation framework for the performance of RD-TENG. In conclusion, ...

Results

Page 4, Line 103-105: As shown in Supplementary Note 1, ..., due to electrostatic shielding, charge cancellation, and reduced interlayer contact-separation efficiency. Inspired by the rattle drum (Supplementary Note 2) ...

Page 4, Line 125-129: ... This forms another charge output channel opposite to the charge-balancing channel of electrode 2. Microscopically, the internal short-circuit electrode enables charge diversion, preventing cancellation from reverse flows in shared paths. Macroscopically, it transforms each external electrode from single-electrode mode into contact-separation mode

by pairing it with vibrating short-circuit electrodes, enabling mode transformation and resolving the electrostatic shielding (Supplementary Notes 3 and 5) ...

Page 9-10, Line 233-236: ...At this stage, a fully independent RD-TENG is fabricated (Supplementary Fig. 19), enabled by the charge dispatch strategy and structural innovations, and **establishes a novel and feasible TENG mode—three-electrode stacked FCS-TENG**. The discussion on the scalability of each strategy can be found in Note S8 (ESI†) ...

Page 10-11, Line 267-272: ...The outputs of the 1-2, 2-3, and rectifier electrode pairs are shown in Supplementary Fig. 23 and Table 2. The above qualitative simulation results and experimental measurements are consistent with the calculations derived from our constructed equivalent capacitance circuit model. It is worth noting that although Structures 1, 2, and 3 exhibit observable charge transfer outputs, the corresponding quantitative model predicts zero output. This discrepancy arises from the non-ideal conditions during actual testing, where experimental errors account for the deviation. Structure 4 achieves a significant output improvement

For clarity, major additions related to Comments #2–#6 are summarized below, with full details provided in later sections:

1. Supplementary Note 1 (**Page 8-10**) clarifies performance limitations in conventional dense-stacked TENGs.
2. Supplementary Notes 2–3 (**Page 11-13**) trace the conceptual and technical evolution from rattle-drum dynamics to RD-TENG's architecture and charge transfer mechanism.
3. Supplementary Note 4 (**Page 14-15**) systematically compares our three-electrode system with prior designs, highlighting key differences.
4. Supplementary Note 5 (**Page 16-25**) introduces a capacitive-network model to quantify the dual-loop charge dispatch mechanism (Fig. 1g–h).
5. Supplementary Note 6 (**Page 26-27**) and Fig. 1i establish a comprehensive performance framework centered on TSD, integrating structure, output, and application relevance.

2. The innovation lies in the specific structural design (rattle-drum mechanics, alternating coatings, push-pin connections) and the particular way the three electrodes are configured and interact—especially the role of the short-circuited internal electrode 2 as a "charge dispatch channel". This is a refinement and specific application of existing concepts, characteristic of an incremental advancement. The authors themselves frame their work as combining "alternate

film coating and a three-electrode design" as well. The "rattle drum inspiration" appears more metaphorical than mechanistically meaningful, serving primarily as branding rather than providing genuine biomimetic insights.

Response: We sincerely thank the reviewer for your careful reading and insightful comments, particularly regarding the fundamental nature of our innovation and the depth of mechanistic understanding. Your feedback prompted us to further clarify the conceptual core of this work and systematically reflect on the current role and evolution of three-electrode designs in TENG research. This has greatly helped us to better articulate the originality of our approach, the underlying physical distinctions, and the mechanistic implications embedded in the naming and design of RD-TENG.

1. Comparison between our three-electrode strategy and previous designs

1) RD-TENG introduces a short-circuit TENG as an internal electrode and alternate film coating to enable charge dispatch.

We appreciate the reviewer's concern about whether our three-electrode structure represents a mechanistic breakthrough or merely a structural modification. In response, we clarify that the RD-TENG's third electrode is not a passive output terminal, as seen in previous designs, but functions as a dynamic short-circuit element enabling internal charge dispatch. Coupled with alternating surface coatings, it forms an internal TENG loop that actively redistributes charge across stacked layers. This establishes a dual-loop system—internal redistribution and external output—thereby:

- i. Charge path diversion to mitigate phase cancellation.
- ii. Mode transformation to suppress electrostatic shielding.

2) Originality and uniqueness of this work compared to previous three-electrode designs

To clarify the mechanistic novelty of our three-electrode design, we reviewed representative literature (**Table N1 in Supplementary Note 4**). While "three-electrode TENGs" have appeared occasionally, most adopt added electrodes for multi-directional sensing or hybrid output, without introducing internal charge redistribution or altering the working mode. These designs remain structurally diverse but mechanistically conventional—still based on the classic two-electrode logic. In contrast, RD-TENG embeds a short-circuited vibrating electrode within a high-density stack FCS-TENG, enabling dynamic charge dispatch and neutralization during operation. This suppresses electrostatic shielding and charge cancellation, key limitations in densely stacked TENGs. The core innovation is not the extra electrode itself, but the reconfiguration of charge paths and working mode—marking a fundamental shift in TENG architecture.

2. Response to the Question on the “Rattle Drum Inspiration” as a Rhetorical Metaphor: From Structure to Dynamics to Assembly: A Fully Integrated Mechanistic Pathway Inspired by the Rattle Drum

We thank the reviewer for raising concerns about whether the “rattle drum inspiration” is purely metaphorical. We clarify that the rattle drum serves as a genuine mechanistic inspiration across the full design pathway—from structural layout to dynamic behavior.

1) Structural Prototype Inspiration:

As shown in Fig. R#1–3 and Fig. S4, the rattle drum’s dual-sided, symmetric percussion inspired the core sandwich structure of RD-TENG: fixed electrode – vibrating electrode – fixed electrode. The vibrating sheet mimics the drum's central striker, under excitation, performs harmonic motion (Fig. S5), alternately contacting the two fixed electrodes. This layout forms the physical and functional basis for the three-electrode charge dispatch system.

Fig. R#1-3. Inspiration mapping from the rattle-drum to the prototype TENG and further to the RD-TENG.

Fig. S4. Vibrating sheet style, parameters, model, and physical image.

Fig. S5. Vibration simulation of the vibrating steel sheet.

2) Inspiration from Rattle Drum Dynamics to Electrodynamics Behavior in RD-TENG

The dynamic principle of the rattle drum—centrifugal force equilibrium and angular momentum conservation, directly inspires the electrodynamic design of the RD-TENG—charge dispatch and potential field equilibrium. In this system, the short-circuited electrode serves both as a conductive pathway and a dynamic field balancer, enhancing electrostatic induction while mitigating shielding effects. This dual-loop configuration facilitates efficient internal charge migration, significantly boosting volumetric power density. Hence, the rattle drum functions not merely as a symbolic reference but as a mechanistic prototype for dynamic to electrodynamic integration.

3) Inspiration for Modular Assembly.

The rattle drum's symmetrical, repeatable, and modular structure inspired the RD-TENG's modular design. Internally, triboelectric units are assembled using a looping mortise-and-tenon method with post-based ring gaskets, allowing rapid, tool-free interlocking without adhesives. This enhances fabrication efficiency, mechanical stability, and scalability.

The analysis of the fundamental differences between our three-electrode design and prior configurations has been added in **Note S4 ESI†**, while the rattle drum–inspired design pathway has been detailed and clarified in **Notes S2–S3 ESI†**.

Revision in *Main Manuscript*:

Page 4, Line 113-115: ...As shown in Fig. 1b and Supplementary Note 3, we proposed a three-electrode system that is fundamentally distinct from conventional three-electrode configurations (comparative analysis in Supplementary Note 4).⁵¹⁻⁵⁴ ...

Page 4, Line 105-106: ...Inspired by the rattle drum (Supplementary Note 2), ...

Page 4, Line 125-129: ...This forms another charge output channel opposite to the charge balancing channel of electrode 2. Microscopically, the internal short-circuit electrode enables charge diversion, preventing cancellation from reverse flows in shared paths. Macroscopically, it transforms each external electrode from single-electrode mode into contact-separation mode by pairing it with vibrating short-circuit electrodes, enabling mode transformation and resolving the electrostatic shielding (Supplementary Notes 3 and 5). ...

References

51. Feng, L. et al. Hybridized nanogenerator based on honeycomb-like three electrodes for efficient ocean wave energy harvesting. *Nano Energy* 47, 217–223 (2018).
52. Liu, X. et al. High-performing honeycomb-structured triboelectric nanogenerator enhanced by triple electrodes for utilizing wind power. *Nano Energy* 118, 108961 (2023).
53. Yang, L. et al. An electrode-grounded droplet-based electricity generator (EG-DEG) for liquid motion monitoring. *Adv. Funct. Mater.* 33, 2302147 (2023).
54. The output improvement of droplet-based nanogenerators through enhanced coupling displacement and conducting currents. *Nano Energy* 121, 109191 (2024).

Revision in *Supplementary Information*:

Note S4. Essential differences between the three-electrode RD-TENG and previous works.

Page 14-15: In prior literature, most TENGs described as “three-electrode structures” typically feature a third electrode functioning as an auxiliary signal extraction layer or sensing terminal, primarily aimed at enhancing directional sensitivity, expanding the detection area, or increasing

the number of output channels. However, although such designs appear to incorporate three electrodes geometrically, they do not establish a functional mechanism for charge path reconstruction.... (text omitted)

Table N1. Comparison between previously reported *three-electrode* configurations and the approach presented in this work

No.	Yr.	newly illustrated schematic	Mode	Function of the Three-Electrode Configuration	Essence	Significance	Assessment	Ref.
1	2018		FS	Enhances energy harvesting density and angle coverage	2-Elect.	Alternating arrangement forms electrodes A, B, and C to collect energy from multiple directions	Merely a geometric addition of electrodes	51
2	2023		SE	Enhances energy harvesting density and angle coverage	1-Elect.	Three independent electrode groups (A, B, C) are formed by dividing the inner aluminum foils symmetrically at 120°, connected in a three-phase circuit	A geometric partitioning to collect outputs from different orientations simultaneously	52
3	2023		SL	Functions like a grating electrode for electrical signal monitoring	2-Elect.	Uses time delays between multiple electrodes to monitor droplet velocity, offering an integrated solution for fluid energy harvesting and intelligent sensing	Adds electrodes in-plane to establish signal-space relationships	53
4	2023		SL	Collects positive and negative charges within droplets	2-Elect.	The top and bottom electrodes separately collect different polar charges; the rear electrode is grounded to improve charge extraction efficiency and output stability compared to conventional droplet-based TENGs	Uses spatially distributed electrodes to extract bipolar charges from droplets	54
6	This work		Stacked FCS	Enables charge dispatch, mode conversion, and output participation	3-Elect.	The three-electrode design constructs dual-loop circuits to mitigate charge cancellation caused by shared electrodes in conventional models, and uses a short-circuit channel to convert traditional CE-type units into CS-type units, addressing intrinsic electrostatic shielding problems	The internal electrode actively participates in both charge dispatch and output, with an irreplaceable role beyond mere geometric addition	This work

Caption: The meanings of the relevant abbreviations are as follows:

SL: freestanding sliding mode; SE: single-electrode mode; SL: solid-liquid contact mode; Stacked FCS: freestanding contact-separation high-density stacked mode. Elect.: electrode

Page 12-13: Note S3. From the rattle-drum-model TENG to the RD-TENG.

Fig. N5a and b illustrate the correspondence of motion states throughout a full operating cycle from the rattle-drum-model TENG to the RD-TENG. At multiple levels... (text omitted). Based on this, a three-electrode TENG dynamic charge dispatch mechanism system is developed... (text omitted) ...the three electrodes on the device are shown in Fig. N6.

Fig. N5. State correspondence diagram from the rattle-drum-type TENG to the RD-TENG

As illustrated in Fig. N5b, the charge transfer mechanism. On the microscopic level: The internal short-circuit electrode's charge circulation promotes the external electrode's charge outer circulation ... (text omitted) On the macroscopic level: The introduction... (text omitted)

Page 11: Note S2. Inspiration from the rattle drum to the prototype device.

As illustrated in Fig. N4a and b, the prototype TENG device is inspired by the mechanical and dynamical features of a traditional rattle drum. ... (text omitted). Structurally, ... (text omitted). Electrically, ... (text omitted). As a result, the inherent electrostatic shielding effect of single-electrode structures—especially problematic in high-density stacks—is significantly mitigated ... (text omitted).

3. The emphasis on triboelectric surface density (TSD) as a primary performance indicator is scientifically questionable. Defining TSD as surface area per unit volume creates a metric easily manipulated through geometric optimization rather than fundamental material or physical improvements. This appears to be metric engineering rather than genuine performance enhancement. The claimed "record" TSD of 2.76 cm^{-1} may simply reflect device miniaturization rather than superior energy conversion efficiency.

Response: We sincerely thank the reviewer for raising the important concern regarding the validity of triboelectric surface density (TSD) as a performance metric. Your insight that TSD

may reflect structural rather than intrinsic physical efficiency has prompted us to carefully re-examine its role. We agree that no single metric fully captures TENG performance. TSD was proposed not to replace conventional electrical indicators, but to complement them—quantifying compact integration, which is crucial for practical deployment. To address your concern, we clarified the scope and limitations of TSD and emphasized its position within a broader, multi-parameter evaluation framework that balances structural metrics and electrical outputs. This approach ensures physical relevance and meaningful comparability across diverse TENG architectures.

1. Limitations of existing performance metrics: incomplete evaluations

As shown in Table R#1-1. As TENGs evolve in structure and application, evaluating their performance using only traditional metrics—like open-circuit voltage, short-circuit current, surface charge density (SCD), or power density—proves insufficient. These parameters, while physically meaningful, are highly sensitive to structural scale, material choices, operating modes, and fabrication variations. More critically, they often fail to reflect mechanical configuration, spatial efficiency, lightweight, cost-effectiveness, or deployment feasibility. For instance, SCD primarily characterizes material-level charge generation but does not account for key structural elements like electrode layout and substrate support etc.

Table R#1-1 Performance indicators used in previous works.

No.	Ref.	Content	Metrics
1	Nat. Commun. 2022, 13 (1)	proposed the concept of intrinsic surface charge density to quantify the theoretical output limits of different TENG materials.	SCD
2	Nat. Commun. 2021, 12 (1), 5470	introduced the surface instantaneous power density as a performance indicator.	SPD (surface power density)
3	Energy Environ. Sci. 2025, 18 (10), 4893–4904	evaluated TENG performance through surface charge density and energy density metrics.	SCD
4	Nat. Commun. 2015, 6 (1), 8376	established a set of standard figures of merit for quantifying TENG performance from both structural and material perspectives;	material and structural parameters
5	Research 2023, 6, 237	emphasized VPD as a key metric and explored how frequency enhancement mechanisms affect VPD.	Peak VPD (volume power density)
6	Nano Energy 2023, 108818	This paper uses VPD, charging rate improvement as key metrics to evaluate the outstanding performance of the M-TEHG in energy harvesting and vibration sensing.	Peak VPD Capacitor charging rate
7	Nat. Commun. 2023, 14, 1023	This paper uses wear resistance and peak SCD as key metrics to evaluate the outstanding performance of metallic glass-based TENGs in achieving high charge generation and durability.	SPD, durability

2. Construction and Requirements of a Comprehensive Evaluation Framework

To enable fair comparison across diverse TENG designs and to support standardized future development, it is imperative to construct a unified, multidimensional evaluation system with clear structural logic and engineering relevance. In Adv. Mater. Technol. 2025, e00184, we

proposed and preliminarily established a comprehensive performance evaluation framework. In this work is presented in Supplementary Note 6 presents its constituent components, physical significance, and the theoretical basis for introducing the core structural metrics of TSD, which serve as a key structural descriptor within the overall system.

3. Role and Significance of TSD in the Comprehensive Evaluation Framework

While TSD is geometry-dependent, it is not trivially optimized. High TSD brings real challenges—like reduced separation efficiency and stronger electrostatic shielding—making it physically meaningful rather than arbitrary. TSD, defined as the total contact area per unit volume (Adv. Mater. Technol. 2025, e00184), was introduced to fill the lack of structural metrics in TENG research. It reflects structural integration and spatial efficiency, especially in multilayer or 3D-stacked systems, offering a volumetric perspective beyond traditional planar performance metrics. It is worth noting that in this study, TSD is not isolated from other indicators but intrinsically linked with them. These correlations—emerging as a result of overcoming structural and output bottlenecks—are illustrated in the updated Fig. 1i and further expanded in Fig. N11 and Note S6 ESI†.

1) Motivation: the demand for structural compactness in distributed energy systems.

2) Physical Significance of high-TSD. i) Spatial Efficiency: High TSD enables denser energy units within compact volumes, ideal for integration into buoys, flexible surfaces, and MEMS. **ii) Array Scalability:** Dense units allow efficient TENG arrays on limited surfaces, enhancing total energy capture and module integration. **iii) Modular Manufacturing:** Compact, loop-assembled designs support standardization and automated fabrication. **iv) Co-Optimization:** Densification introduces challenges (e.g., shielding, cancellation), prompting a shift to coupled structural-electrical-mechanical design. In short, high-TSD is not just about output density—it is key to future scalable, modular TENG systems.

3) Scientific Applicability and Constraints. We agree that TSD must not become “metric engineering.” TSD is not linearly linked to performance—excessive densification leads to shielding, charge cancellation, and reduced separation, as detailed in Note S1. Thus, TSD has a physical upper limit. Its scientific value lies in balancing structure and performance. In this work, we overcome high-TSD bottlenecks via a three-electrode dispatch mechanism and short-circuit loop, ensuring structural gains yield real output improvements.

4. Comprehensive Evaluation Framework and in This Work

Importantly, TSD is not used as a standalone metric in this study. We recognize that insufficient emphasis in the original manuscript may have led to the reviewer overlooking the presence of

additional performance indicators.

1) The original manuscript evaluates the work from: The first dimension highlights the performance gap between the proposed RD-TENG and three types of conventional stacked FCS-TENG structures, differentiated solely by the presence or absence of the charge dispatch strategy, as shown in Fig. 2c–e.

Fig. 2c–e Comparison of charge, voltage, and current outputs at varying frequencies with a 70mm amplitude between the RD-TENG 1-3 electrode pair and traditional models.

The original Fig. 1 (h) Comparison of the maximum peak power density with other works. (i) Comparison of the structural characterization parameters with other works.

Beyond structural metrics, RD-TENG was benchmarked across four dimensions—VPD, VCD, TSD, and SSA (The original Fig. 1h–i).

2) In the revised manuscript, we introduced the SCD indicator, and the evaluation framework was updated as follows:

- a) **The Record TSD:** 2.76 cm^{-1} , among the highest reported.
- b) **Lightweight design:** 345 g total weight, $\text{SSA} = 2.90 \text{ cm}^2 \text{ g}^{-1}$.
- c) **Enhanced output:** $\text{VPD} = 136.74 \text{ W} \cdot \text{m}^{-3}$; $\text{VCD} = 11.69 \text{ mC} \cdot \text{m}^{-3}$; $\text{SCD} = 42.2 \text{ } \mu\text{C} \cdot \text{m}^{-2}$.
- d) **System robustness:** 12,760 cycles without decay;
- e) **Energy storage capability:** 47 μF capacitor charged to 9.7 V in 1 min.
- f) **Scalable & adaptable:** Lego-style modular array integration and real-sea wave harvesting.

The five-indicator illustration is shown in the revised Figure 1k–l.

Finally, we deeply appreciate the reviewer’s scientifically grounded challenge to the TENG evaluation methodology. Your comment prompted us to reflect critically on how to avoid “metric inflation” and instead build a more generalizable and scientifically robust performance framework. In the revised manuscript, we have clarified this through updated figures and explanations, with additional elaboration provided in Note S6 ESI†. Furthermore, we now explicitly define the scope and limitations of TSD—positioning it not as an absolute performance target, but as one dimension within a multi-indicator structural evaluation system—thus avoiding its misinterpretation as a singular figure of merit.

Revision in Main Manuscript:

Fig. 1(i) Correlation mapping between structural indicators and electrical performance metrics of RD-TENG under the comprehensive evaluation framework.

Fig. 1(k) Comparison of the maximum VPD with other works. **(l)** Comparison of the structural characterization parameters with other works.

Page 2-3, Line 53-65: As such, ... In addition, when combined with the Specific Surface Area (SSA),¹⁷ the triboelectric surface area per unit mass—it enables evaluation of the device's lightweight design. Together, TSD and SSA constitute key structural performance metrics for TENG devices. Improving TSD is expected to overcome current technical bottlenecks and accelerate TENG industrialization. Compactness, triboelectric layer integration, and lightweight design are regarded as future trends for distributed energy devices.¹⁷ However, as TSD and triboelectric layer density increase, achieving a sufficient contact-separation distance becomes increasingly challenging. Furthermore, the traditional structurally decomposed fundamental unit is the single-electrode mode, which inherently suffers from electrostatic shielding effects;³¹ resulting in device internal electrostatic shielding intensifies with increasing layer density, and in free-standing vertical contact-separation (FCS) configurations, the use of shared electrode layers leads to severe charge cancellation between stacked units, further limiting the overall electrical output.

Page 3, Line 87-89: ... Compared to similar devices, the five-in-one evaluation system, including structural and electrical performance indicators, comprehensively surpasses previous results (Fig. 1k-1l).³²⁻⁴⁶ ...

Page 7, Line 186-192: To rigorously evaluate the RD-TENG's performance, we established a comprehensive five-dimensional assessment framework (Supplementary Note 6), including structural metrics such as TSD and SSA, and electrical performance metrics such as VPD, volumetric charge density (VCD), and surface charge density (SCD). The correlation among these metrics is illustrated in Fig. 1i. Under the charge dispatch strategy and structural innovations, improvements in structural metrics directly enhance electrical performance. To better clarify the device performance, additional metrics for array modularity, scenario adaptability, and durability were introduced, together forming the comprehensive evaluation framework of the RD-TENG.¹⁷

Page 8, Line 200: ... the VCD reaches 11.69 mC m^{-3} , SCD reaches $42.23 \text{ } \mu\text{C m}^{-2}$...

Page 21, Line 550-551: ... Based on these results, we further established a comprehensive evaluation framework for the performance of RD-TENG ...

17. Tang, W., Liu, G. & Wang, Z. L. Water-wave driven triboelectric nanogenerator networks: a decade of March in blue energy and beyond. *Adv. Mater. Technol.* 2025, e00184.

Revision in *Supplementary Information: Supplementary Note 6:*

Page 26-27: Note S6. Construction of a comprehensive performance evaluation system for TENG systems

Under the ongoing evolution of TENG architectures and the continuous expansion of their application scenarios, establishing a systematic performance evaluation framework that integrates structural universality with electro-physical insight has become one of the core challenges in the field... (text omitted)

Table N2: Comprehensive evaluation metric.

No.	Attribute	Indicator	Definition	Significance
1	Structural Performance Metrics	TSD	Defining as the triboelectric surface area per unit device volume	Quantifies the integration level and compactness of energy-harvesting layers within the device. It serves as a measure of both spatial efficiency and material utilization, reflecting the extent to which a structure maximizes its internal active layer density.
		SSA	Defined as the triboelectric surface area per unit device mass	SSA reflects the design's lightweight efficiency and cost-effectiveness. It is particularly important in portable, wearable, or large-area deployable applications where weight and material economy are critical.
2	Electrical Performance Metrics:	VCD	Corresponding peak output per unit volume	Reflects the combined effects of internal structural integration and material-coupling efficiency among triboelectric layers, serving as measure linking electrical and structural performance. reflecting the spatial energy conversion capacity and practical output delivery potential of the device.
		VPD		
		SCD SPD	Corresponding peak output per unit area	Surface Charge Density (SCD) and Surface Power Density (SPD): These are often material-dominated metrics, reflecting the intrinsic charge-generation capability of dielectric materials and the influence of surface engineering (e.g., nanostructuring, charge injection).
3	Application-Oriented Indicators		These metrics assess the system's readiness for real-world deployment	Manufacturability and packaging compatibility; Energy storage efficiency and charging rates; Load-driving capabilities; Mechanical robustness, durability under cyclic stress; Response uniformity in large-scale arrays; Scenario adaptability across diverse environmental and excitation conditions.

From a long-term and industry-oriented perspective, compact structures, dense integration of energy-harvesting units, and lightweight miniaturization represent key development directions for TENGs in distributed energy and sensing applications. ... (text omitted)

4. The comparisons with "traditional models" lack fairness. The paper does not adequately demonstrate that compared devices represent state-of-the-art optimized designs. Previous literature shows charge densities reaching $1250 \text{ } \mu\text{C m}^{-2}$ (<https://doi.org/10.1038/s41467-022-33766-z>) in controlled conditions and 5.4 mC m^{-2} for DC-TENGs

(<https://doi.org/10.1038/s41467-020-20045-y>), suggesting the performance improvements claimed here are not as exceptional as presented. Without a clear presentation and comparison of the RD-TENG's intrinsic surface charge density, it is difficult to ascertain whether the reported high volumetric outputs stem from a fundamental breakthrough in charge generation/separation or primarily from the device's successful geometric densification. The authors should provide this data and discussion.

Response: We sincerely thank the reviewer for raising the crucial issue of comparative fairness and SCD (surface charge density) relevance, which addresses a core challenge in TENG evaluation. We fully acknowledge the benchmark works you cited ($1250 \mu\text{C}\cdot\text{m}^{-2}$, $5.4 \text{mC}\cdot\text{m}^{-2}$) and their influence on our development. Your comment prompted deeper reflection on the true source of RD-TENG's performance gains and its position within the TENG field. Below, we respond from multiple dimensions.

1. Significance and limitations of the two referenced studies

1) The research significance of the above two works.

We fully agree that the cited works are TENG milestones:

- i) Ref. 1 ($1250 \mu\text{C}\cdot\text{m}^{-2}$) defined intrinsic SCD limits under vacuum, eliminating air breakdown.
- ii) Ref. 2 ($5.4 \text{mC}\cdot\text{m}^{-2}$) achieved high output via corona discharge in sliding DC-TENGs.

2) Limitations and practical considerations.

However, both rely on highly controlled conditions:

- i) Vacuum or sustained discharge limits real-world deployment.
- ii) Sliding-mode TENGs face strong wear, friction, and require a high driving force—unsuited for weak excitations and long-term operation. Similar works on high-SCD DC-TENGs that struggle to balance high output with practical deployment are summarized in Table R#1-2.

RD-TENG takes a mechanistically distinct, application-oriented approach, addressing electrostatic shielding in stacked systems via a three-electrode charge dispatch design. RD-TENG durability test demonstrates that it maintains stable output over 12,760 cycles (Fig. S34, ESI[†]), with no observable degradation—underscoring its potential for long-term applications.

Table R#1-2 Other high-output DC-TENGs

Article (DC-TENG)	Motion type of TENG	Output charge	Charge density (mC m^{-2})
Sci. Adv., 2019, 5, eaav6437.	Sliding	157nC	0.43
Nat. Commun., 2021, 12, 4686.	Sliding	4400nC	8.8
Nat. Commun., 2020, 11, 6186	Sliding	2700nC	5.4
Adv. Energy Mater., 2022, 12, 2200963.	Sliding	1500nC	0.4

Article (DC-TENG)	Motion type of TENG	Output charge	Charge density (mC m^{-2})
Energy Environ. Sci. 2024, 17 (24), 9590–9600.	Rotation Sliding	115.94 μC	7.3
Energy Environ. Sci. 2024, 17 (2), 580–590.	Rolling Sliding	244.2 μC	10.06

Fig. S34. Durability test of the 1-3 electrode pair of RD-TENG based on low-frequency and amplitude-amplifier of magnetic repulsive pendulum under linear excitation.

3) Limitations of single-parameter evaluation.

While SCD reflects the triboelectric potential of materials, it does not capture key factors affecting real-world TENG performance—such as device architecture, mechanical durability, spatial efficiency, or environmental adaptability. TENG output depends not only on material properties but also on structural dynamics and excitation conditions. Thus, SCD alone cannot assess system-level readiness or practical deployment value.

2. Comparative framework in the original manuscript of this work.

The relevant description has been provided in page 15-16 of our response to your Comment #3.

3. Updated SCD and Multi-Metric Evaluation in This Work.

Similarly, the relevant description has been provided in page 15-16 of our response to your Comment #3. (We recalculated RD-TENG’s surface charge density (SCD) as $\sim 42.23 \mu\text{C m}^{-2}$ under ambient, contact–separation conditions—Although it is lower than the theoretical value under ideal and controlled conditions, it surpasses that of existing high-density layered TENGs.)

We sincerely thank you for prompting us to improve both data completeness and theoretical rigor. Your comments have been instrumental in elevating the scientific clarity, comparability, and field relevance of our work. In the revised manuscript, we emphasize the significance of RD-TENG in delivering “performance-coordinated design under practical operating conditions,” and have added the relevant unit conversions and expanded discussion to fully address your concerns.

Revision in Main Manuscript: Fig. 11 (Page 17 of this response manuscript) and:

Page 8, Line 200: ... the VCD reaches 11.69 mC m^{-3} , SCD reaches $42.23 \mu\text{C m}^{-2}$

Page 17, Line 434: ...with a VCD of 11.69 mC m^{-3} and a SCD of $42.23 \mu\text{C m}^{-2}$) ...

Page 21, Line 547: ... achieving a VCD of 11.69 mC m^{-3} and SCD of $42.23 \mu\text{C m}^{-2}$...

Revision in Supplementary Information: Page 6:

Table 1. Comparison of the RD-TENG with other works in terms of unit volume output charge density, specific surface area density, friction layer area-to-mass ratio, and peak power density.

Type	Reference	Charge (μC)	Friction area (cm^2)	Volume bulk (cm^3)	Weight (g)	SCD ($\mu\text{C m}^{-2}$)	VCD (mC m^{-3})	TSD (cm^{-1})	SSA ($\text{cm}^2 \text{g}^{-1}$)	VPD (W m^{-3})
RD-TENG	This work	4.22	999.35	361.73	345	42.23	11.69	2.76	2.9	1-3 electrode pair 136.74 1-2 electrode pair 102.51 2-3 electrode pair 84.97 Rectified electrode pair 114
FH-TENG	Ref. [32] 2024	7.9	2052.75	769.69	621.2	38.48	10.26	2.67	3.3	14.94
OM-TENG	Ref. [33] 2024	60.82	15226.63	8651.49	7110	39.94	7.03	1.76	2.14	28.9
SO-TENG	Ref. [34] 2024	1.5	900	1359.78	800	16.67	1.1	0.66	1.13	4.44
D-Z-TENG	Ref. [35] 2024	1.2	360	267	500	33.33	4.49	1.35	0.72	55.4
DA-TENG	Ref. [36] 2024	0.62	343	/	/	18.08	0.19	0.8	/	7.51
GA-TENG	Ref. [37] 2024	/	/	/	/	/	/	/	/	20.4
LI-TENGs	Ref. [38] 2023	0.76	216	108	115.6	35.19	7	2	1.87	48.47
T-TENG	Ref. [39] 2023	1.15	460.86	430.71	358	24.95	2.67	1.07	1.29	18.9
HM-TENG	Ref. [40] 2022	2.4	951.72	924	742.6	25.22	2.6	1.03	1.28	2.44
O-TENG	Ref. [41] 2022	29.93	12160	6170.73	7300	24.61	4.85	1.97	1.67	1.62
MH-TENG	Ref. [42] 2023	2.9	783.78	698.28	353.25	37.00	4.15	1.12	2.2	23.2
D-TENG	Ref. [43] 2022	0.87	480	5400	280.8	18.13	0.16	0.09	1.71	6.35
S-TENG	Ref. [44] 2022	0.16	62.8	448	121.1	25.48	0.36	0.14	0.52	7.39
F-TENG	Ref. [45] 2022	0.18	65	2574.26	80.96	27.69	0.07	0.03	0.8	16.96
SR-TENG	Ref. [46] 2022	0.15	202.3	736.63	281	7.41	0.2	0.27	0.72	15.4

5. The COMSOL simulations of electric potential provide a qualitative visualization of the proposed charge dispatch mechanism. However, for a significant advance, a more comprehensive theoretical framework is expected. This would ideally include rigorous electrostatic analysis that quantitatively predicts the reduction in electrostatic shielding and the enhancement in potential difference due to the specific three-electrode configuration and charge dispatch strategy, moving beyond phenomenological descriptions.

Response: We sincerely thank the reviewer for this insightful and professional question. Your emphasis on the need for a quantitative electrical model aligns with a core theoretical challenge of this study. We acknowledge that our original COMSOL simulations were primarily qualitative and lacked rigorous modeling of mechanism–structure coupling. In response, we have now developed a capacitance-network-based framework to quantitatively explain how the three-electrode charge dispatch strategy mitigates shielding and enhances output.

1. Theoretical foundation for RD-TENG model development.

We appreciate the opportunity to clarify the theoretical foundation behind the RD-TENG model.

While classical TENG modes (CS, LS, FS, SE) and their models are well established, the RD-TENG—with its embedded short-circuit unit operating under a freestanding contact-separation (FCS) mode—lies outside these categories. Existing stacked FCS-TENGs face severe electrostatic shielding and charge cancellation, and lack mature theoretical frameworks. However, starting from the theoretical understanding of inherent electrostatic shielding in SE-TENGs, we derive the inspiration for constructing the quantitative analytical model. Previous studies (e.g., Chapter 4 of *TRIBOELECTRIC NANOGENERATORS* and *Adv Funct Mater*, 2014, 24(22):3332-3340) have shown that the shielding effect of the free electrode in SE configurations limits the maximum energy conversion efficiency to approximately 50%. These theoretical foundations provide important support for the proposed charge dispatch strategy and model development, offering predictive insights and mitigation approaches for performance degradation in high-density stacked configurations.

2. Limitations of Conventional Models and Strategy for Overcoming Them

Conventional high-density FCS-stacked TENGs can be deconstructed into SE-mode units, revealing their inherent reliance on SE operation. This leads to three key limitations: **1)** Electrostatic shielding worsens with stacking density. **2)** Shared electrodes cause charge cancellation. **3)** Displacement is reduced, lowering contact-separation efficiency.

To address these, RD-TENG introduces: **1)** Unit-level transition from SE (Fig. N3) to CS mode (Fig. N7); **2)** Dual-loop charge redistribution via internal short-circuit electrodes; **3)** Structural innovations to improve contact dynamics.

Added Fig. N3. Schematic decomposition of the basic unit in the traditional FCS model.

Added Fig. N7. Decomposition diagram of the fundamental unit of the RD-TENG model

3. Theoretical Model Construction

Based on the above analysis, we constructed equivalent capacitive circuit network models for

different structures and, applying Kirchhoff's circuit laws, derived quantitative predictive models for both traditional structures without the charge dispatch strategy and the RD-TENG with charge dispatch. The theoretical analytical results were further visualized and simulated. Comparative analyses between theoretical model results, charge transfer principles, finite element potential simulation results, and experimental measurements revealed strong consistency, confirming that the charge dispatch strategy in RD-TENG effectively resolves charge cancellation and inherent electrostatic shielding through charge diversion and mode transformation. Detailed analysis, derivations, visualizations, and conclusions are provided in Supplementary Note 5 (Page 16-25).

Revision in Main Manuscript:

Added Fig. 1 Equivalent capacitance circuit model: (g) Traditional models, (h) This work

Page 2-3, Line 61-65: However, ... Furthermore, the traditional structurally decomposed fundamental unit is the single-electrode mode, which inherently suffers from electrostatic shielding effects;³¹ resulting in device internal electrostatic shielding intensifies with increasing layer density, and in free-standing vertical contact-separation (FCS) configurations, the use of shared electrode layers leads to severe charge cancellation between stacked units, further limiting the overall electrical output.

Ref. 31. Niu, S. et al. Theoretical investigation and structural optimization of single-electrode triboelectric nanogenerators. *Adv. Funct. Mater.* 24, 3332–3340 (2014).

Page 6-7, Line 161-185: The RD-TENG employs a charge dispatch strategy ... (text omitted) For traditional models, the short-circuit transferred charge Q and open-circuit voltage V are given by:

$$Q = 0$$

$$V = 0$$

For the RD-TENG:

$$Q = \frac{\frac{\sigma_T S}{C_D} \left(\frac{1}{C_{s-x}} - \frac{1}{C_x} \right)}{\left(\frac{1}{C_x} + \frac{1}{C_D} \right) \left(\frac{1}{C_{s-x}} + \frac{1}{C_D} \right)}$$

$$V = \frac{2\sigma_T S}{C_D} \frac{\frac{1}{C_x} - \frac{1}{C_{s-x}}}{\frac{1}{C_x} + \frac{1}{C_{s-x}} + \frac{2}{C_D}}$$

... (text omitted)

Thus, we obtain a quantitative theoretical model for the RD-TENG, demonstrating that the equivalent capacitance configuration effectively overcomes the lack of electrical output encountered in traditional models. Detailed analysis, derivations, and visualizations are presented in Supplementary Note 5.

Revision in *Supplementary Information*:

Page 16-25: (As the derivation process is lengthy, only a brief excerpt is provided here)

Note S5. Quantitative theoretical model.

1. Strategy

Specific pathways by which the charge dispatch strategy and structural innovation strategies overcome the limitations of traditional models can be summarized as follows:

1) **Path diversion:** ... (text omitted)

2) **Mode transformation:** ... (text omitted)

3) **Engineering innovations:** ... (text omitted)

2. For traditional configurations (applicable to Structures 1, 2, and 3):

... (text omitted)

3. For the RD-TENG configuration:

... (text omitted)

As shown in the simulation results, ... (text omitted) Therefore, the results of the quantitative analytical model remain consistent with the charge transfer mechanism, finite element simulation, and experimental measurements. Together with the subsequent discussions in the main text, this work establishes a closed-loop validation framework that spans mechanism analysis, charge transfer principles, quantitative modeling, finite element simulations, and experimental verification—forming a mutually corroborating chain of evidence.

6. The extensive demonstrations of wave energy harvesting, including under real-ocean conditions (Fig. 5), are impressive and showcase the practical potential of the engineered RD-

TENG system. While this applied aspect is a strength, its contribution to the manuscript's suitability for the journal will heavily depend on the perceived significance and fundamental novelty of the core TENG device itself. If the core TENG mechanism is considered a sufficiently innovative advance, the application robustly underscores its impact.

Response: We sincerely appreciate the reviewers' high praise for the wave energy harvesting capability and real-world marine environment testing demonstrated in this work. As pointed out, while application scenarios effectively reflect the device's engineering potential, the true academic value and publication suitability for a journal like *Nature Communications*, which focuses on fundamental scientific breakthroughs, lie in the innovative contributions made at the core mechanism level of the device. We fully acknowledge your point. However, our work is not application-oriented; instead, it aims to address the inherent challenges in high-density stacking through an innovative mechanism. A comprehensive and closed-loop validation framework has been established around the proposed charge dispatch strategy. In the revised manuscript, these validations include:

- 1. Mechanism Innovation Overcoming Electrical Performance Bottlenecks**
- 2. Charge Transfer Principle Analysis and Validation**
- 3. Finite Element Simulation Analysis Verification**
- 4. Theoretical Quantitative Model Construction and Validation.**
- 5. Model Comparison and Experimental Validation**
- 6. Structural Innovation to Match Mechanism to Overcome Mechanical Limits**
- 7. Multiplatform Experimental and Real-Sea Validation**

We sincerely thank the reviewer for the comprehensive and constructive feedback across the full research chain. While this work is exemplified through a specific RD-TENG, its core contribution lies in establishing a generalizable three-electrode charge dispatch framework and a mechanism-oriented design paradigm. This addresses key trade-offs in high-density TENGs and provides a systematic path for structural design, charge regulation, and platform adaptability. We believe this integrated mechanism–structure–system approach offers both engineering value and forward-looking insights for the TENG field. We respectfully hope the reviewer re-evaluates the originality and system-level innovation of this work and considers it for publication in *Nature Communications*.

Revision in *Main Manuscript*:

- 1. In terms of charge transfer principle validation: Page 4, Line 125-129: ...**This forms

another charge output channel opposite to the charge-balancing channel of electrode 2. Microscopically, the internal short-circuit electrode enables charge diversion, preventing cancellation from reverse flows in shared paths. Macroscopically, it transforms each external electrode from single-electrode mode into contact-separation mode by pairing it with vibrating short-circuit electrodes, enabling mode transformation and resolving the electrostatic shielding (Supplementary Note 3) ...

2. In terms of finite element simulation validation: Page 10, Line 248-249: ...To further verify the output differences between RD-TENG (Structure 4) and traditional models (Supplementary Note 1), we conducted a comparative study with three conventional structures that we specifically designed and fabricated. As shown in Fig. 2b, the potential simulation diagrams of the three structures show that electrodes 1 and 3 ...

3. In terms of theoretical quantitative modeling validation: Page 6-7, Line 161-185: ... (text omitted)

Thus, we obtain a quantitative theoretical model for the RD-TENG, demonstrating that the equivalent capacitance configuration effectively overcomes the lack of electrical output encountered in traditional models. Detailed analysis, derivations, and visualizations are presented in Supplementary Note 5.

4. In terms of comparative experimental validation: Page 10-11, Line 267-272: ...The outputs of the 1-2, 2-3, and rectifier electrode pairs are shown in Supplementary Fig. 23 and Table 2. The above qualitative simulation results and experimental measurements are consistent with the calculations derived from our constructed equivalent capacitance circuit model. It is worth noting that although Structures 1, 2, and 3 exhibit observable charge transfer outputs, the corresponding quantitative model predicts zero output. This discrepancy arises from the non-ideal conditions during actual testing, where experimental errors account for the deviation. Structure 4 achieves ...

Revision in Supplementary Information: The newly added and revised content in the Supplementary Information has already been detailed in our responses to Questions 2–5 and will not be repeated here; only the corresponding sections are listed below for reference.

1. In terms of charge transfer principle validation: Page 12-13; Supplementary Note 3

3. In terms of quantitative modeling validation: Page 16-25; Supplementary Note 5

Reviewer #2 (Remarks to the Author):

This work presents a novel triboelectric nanogenerator inspired by the rattle drum (RD-TENG) that can overcome the limitations of output performance due to electrostatic shielding and low displacement amplitude in high density TENGs. In addition, it can enhance the output by raising the triboelectric surface density (TSD). In this design, a three-electrode configuration combined with alternating film coatings, enabling charge dispatch and electrostatic potential balancing is proposed. For better contact-separation, traditional springs is replaced by a laser-etched elastic steel sheet. Also, a push-pin contact system and Lego-like modular locking for easy assembly into arrays are also introduced. Some key achievements are highlighted such as the record TSD of 2.76 cm^{-1} , the output volage is up to 2200 V, and peak power density is 136.74 W m^{-3} , it is 6-fold output versus traditional models.

The manuscript can be accepted after addressing the following comments:

Response: We sincerely thank the reviewer for the thorough evaluation and insightful feedback. Your comprehensive and accurate summary of our work, highlighting innovations in charge dispatch strategy, fabrication methods, structural design, and engineering optimization, greatly affirms the significance of our efforts in enhancing output performance. Your encouraging recognition, coupled with the following professional suggestions, has strengthened our confidence in this research direction and provided valuable guidance for refining the manuscript and expanding our future scope.

Major comments:

1. Is it possible to use the three-electrode charge dispatch strategy for other TENG system design?

Response: We sincerely thank the reviewer for raising this forward-looking, insightful, and valuable question. Rather than focusing solely on the local functional performance of the proposed structure, your comment explores its broader applicability and engineering adaptability across different TENG architectures from a fundamental perspective. Your question has helped us further reflect on the broader utility of charge dispatch strategy in various energy conversion platforms.

The three-electrode charge dispatch strategy not only overcomes output limits in high-density TENGs but also avoids complex materials or structures. Using standard materials and simple design—an internal short-circuited electrode and alternate film coatings—it optimizes charge paths and field distribution. This strategy is extendable to other TENG modes, and our team is exploring its broader application.

1. In vertically freestanding contact-separation (FCS) structures

Our results reveal that traditional dense FCS structures are hindered from achieving high output by motion constraints, electrostatic shielding, and charge cancellation (newly Note S1). RD-TENG overcomes these challenges using alternating film coating and a three-electrode charge dispatch strategy, offering a novel and generalizable approach for multilayer FCS architectures.

2. Beyond FCS, this strategy holds great potential for FS mode.

As shown in Fig. R2#-1, FS-mode (freestanding sliding) TENGs typically underutilize the sliding block, which lacks an electrical function. We propose that by adding a dielectric layer of opposite electronegativity and integrating a short-circuit electrode, the slider can serve as a third electrode for charge dispatch—enhancing energy utilization and output. This concept parallels the charge dispatch mechanism in FCS-mode systems by addressing electrostatic shielding. As this is ongoing work, detailed designs will be shared in future studies.

Fig R2#-1 Traditional FS-TENG.

3. More broadly, in other fundamental TENG working modes.

Such as conventional contact-separation (CS) and in-plane sliding, whenever at least four well-defined electrode layers with relative motion exist, a charge dispatch path can be established by introducing a short-circuited internal electrode through rational film layering and field redistribution. This enables enhanced output and improved spatial charge utilization. The strategy is particularly promising in dense multilayered TENG systems, offering a generalizable route to boost energy conversion efficiency.

Beyond this mechanism, our work emphasizes scalable, application-driven structural strategies, including modular integration, wire-free connections, rapid stacking, and adaptive expansion platforms. These ensure compatibility, replicability, and industrial feasibility, supporting system-level deployment. Details are provided in Chapter 2, Note S8, ESI†.

Revision in *Main Manuscript*:

Page 9-10, Line 233-236: At this stage, a fully independent RD-TENG is fabricated (Supplementary Fig. 19), enabled by the charge dispatch strategy and structural innovations,

and establishes a novel and feasible TENG mode—three-electrode stacked FCS-TENG. The discussion on the scalability of each strategy can be found in Supplementary Note 8

Revision in *Supplementary Information: Chapter 2 of Supplementary Note 8:*

Page 9-10, Line 46-48: (As the content is lengthy, only a partial excerpt is provided here)

Note S8. Discussion on materials, engineering choices, scalability, limitations, and future strategies.

... (text omitted)

2. Strategy scalability.

... (text omitted)

At the device level:

1) Charge dispatch strategy scalability: The proposed three-electrode charge dispatch strategy, ... (text omitted)

Importantly, this strategy is also extendable to the free-standing sliding mode (FS-TENG) and more complex hybrid structures. In FS-mode devices, ... (text omitted) (e.g., CS, LS, RS modes) can benefit from this strategy, ... (text omitted)

2) Vibrating sheets: ... (text omitted)

3) Contact push pin electrode connection: ... (text omitted)

4) Looping stacking method for power units: ... (text omitted)

External scalability aspects:

1) Rotating-clasp-device array assembly: ... (text omitted)

2) Frequency-reducing and amplitude-amplifying magnetic repulsion platform: ... (text omitted)

2. Why were spring steel chosen as the vibrating element, given that it is susceptible to corrosion over time in marine environments?

Response: We sincerely thank the reviewer for raising this critical question regarding the material selection of the vibrating component. Your insight not only focuses on the rationality of choosing core structural materials but also deeply addresses their reliability, corrosion resistance, and long-term feasibility in complex application scenarios such as the marine environment. This is of significant value in guiding the engineering adaptability of our research. In response, we offer the following detailed explanation:

1. Superior performance of spring steel sheets

In this study, we selected Mn65 silicon-manganese spring steel as the vibrating sheet material due to its excellent elasticity, rigidity, conductivity, triboelectric positivity, surface durability, moderate mass for built-in oscillation, and superior laser processability. It effectively integrates structural support, electrode functionality, and triboelectric roles, enabling stable, efficient, and durable performance crucial for RD-TENG operation. The detailed content will be supplemented in Chapter 1 of Supplementary Note 8.

2. Anti-corrosion strategies in marine environments

1) Operation above the water surface: It is important to clarify that the RD-TENG is not a submerged device. It is suspended above the water surface by a buoyant platform and driven by wave-induced motion through the magnetic repulsion pendulum. Therefore, the core power generation unit does not come into direct contact with seawater during operation, significantly reducing corrosion risks for internal materials.

2) Multi-level sealing and waterproofing: To further mitigate environmental exposure, the device incorporates a robust set of protective measures: **i. Fully enclosed casing:** The outer shell is fabricated using high-density 3D-printed PLA, combined with top and bottom acrylic sealing plates to form a rigid, enclosed chamber. **ii. Industrial-grade sealing at all interfaces:** Critical seams (electrode ports, rotating clasp joints, conductive through-holes) are sealed with industrial-grade waterproof silicone (e.g., Dow Corning 732), cured for 24 hours to ensure airtight integrity. **iii. Internal structural buffering:** The internal components are layered in a stacked, overlapping manner, with the outermost sheets providing an added layer of protection, buffering internal parts from external stress and potential ingress. The detailed content will be supplemented in Chapter 3 of Supplementary Note 8.

3. Reasons for Not Choosing Stainless Steel

Marine-grade stainless steel was not selected primarily due to its lower elastic modulus, weaker resilience, inferior conductivity, and poorer machinability compared to Mn65 steel. These drawbacks would compromise elastic response, output performance, and manufacturability, hindering large-scale integration. In contrast, Mn65 steel achieves an optimal balance of mechanical, electrical, and structural properties, with corrosion effectively managed through encapsulation, making it a superior engineering choice.

In summary, Mn65 spring steel combines excellent mechanical performance, structural integration, and fabrication compatibility, making it highly suitable for RD-TENG vibrating sheets. Its effectiveness has been validated in multiple studies across diverse TENG architectures and scales (e.g., Adv. Energy Mater. 2025, 15, 2402781, Adv. Funct. Mater. 2024,

34, 2406775, *Adv. Sci.* 2022, 9 (35), 2204407), demonstrating strong reproducibility and scalability. While current encapsulation provides sufficient protection for surface-level marine use, we acknowledge limitations under long-term harsh conditions. Future work will explore anti-corrosion coatings, full encapsulation, and alternative materials to enhance durability. We thank the reviewer for this important suggestion, which has been addressed in the revised manuscript and Note S8 of the ESI†.

Revision in Supplementary Information: Chapter 1 of Supplementary Note 8:

Page 42-43: (As the content is lengthy, only a partial excerpt is provided here)

Note S8. Discussion on materials, engineering choices, scalability, limitations, and future strategies.

1. Selection of core materials and engineering design strategies:

(1) Mn65 silicon-manganese spring steel

... (text omitted)

- 1) Outstanding elastic modulus and structural rigidity: ... (text omitted)
- 2) Good electrical conductivity and positive triboelectric polarity: ... (text omitted)
- 3) Excellent fatigue resistance and surface stability: ... (text omitted)
- 4) Intrinsic mass-loading characteristic: ... (text omitted)
- 5) Superior machinability and structural design adaptability: ... (text omitted)

Table N15. Material parameters of Mn65 spring steel (based on Chinese National Standard GB/T 1222-2016, ASTM A689, and the Mechanical Engineering Materials Handbook)

Category	Property	Value / Range	Unit	Notes
Chemical Composition	Carbon (C)	0.62–0.70	wt %	Primary strengthening element
	Manganese (Mn)	0.90–1.20	wt%	Improves hardenability
	Silicon (Si)	0.17–0.37	wt%	Solid-solution strengthening
	Phosphorus (P)	≤0.035	wt%	Controlled impurity
	Sulfur (S)	≤0.035	wt%	Controlled impurity
Mechanical Properties	Ultimate tensile strength (σ_u)	≥980 (up to 1200–1600 after cold rolling)	MPa	Dependent on heat treatment
	Yield strength (σ_y)	≥785	MPa	—
	Young's modulus (E)	~200	GPa	Typical for steels
	Elongation at break (δ)	≥8 (annealed state)	%	—
	Hardness (annealed)	≤ HB 285 (up to HRC 40–50 when cold-rolled)	—	—
Heat Treatment Parameters	Quenching temperature	830–860	°C	Oil quenching
	Tempering temperature	400–500	°C	Balances strength and toughness
	Annealing temperature	680–720	°C	Softened by controlled cooling
Spring Performance	Fatigue limit (10^7 cycles)	450–600	MPa	Can be improved by 20%–30% via shot peening
	Service temperature range	–40 to +200	°C	Derating is required at elevated temperatures
Physical Properties	Density	7.85	g cm ⁻³	—
	Coefficient of thermal expansion	11.5×10^{-6}	/°C	In the range of 20–100 °C
	Thermal conductivity	~50	W/ (m K)	—

3. Potential limitations and future mitigation strategies.

... (text omitted)

Second, corrosion remains a critical challenge in marine deployments. While this work has implemented multiple layers of engineering protection—structurally, ... (text omitted) However, long-term stability under extreme conditions—such as prolonged salt spray, UV exposure, and turbulence—requires further empirical validation. ... (text omitted) Future work will explore surface treatments (e.g., electroplating, epoxy coatings), enhanced module sealing, the use of elastic composites or corrosion-resistant alloys, and systematic evaluation through corrosion testing and simulation. ... (text omitted)

3. Please prove the stable charge transfer and the performance of RD-TENG after long-term operation?

Response: We sincerely thank the reviewer for highlighting the durability concern, which touches on a fundamental challenge in translating TENGs to real-world applications. Long-term output stability is essential for both device reliability and sustainable energy harvesting. We will explain the stable output of the RD-TENG from the following four perspectives:

1. Mechanism Level:

As detailed in Note S1-5 ESI†, RD-TENG adopts a three-electrode charge dispatch design that avoids charge cancellation from shared electrodes and converts the base unit to a vertical contact-separation mode, effectively suppressing electrostatic shielding and dielectric breakdown. As a result, it ensures stable, sustained output.

2. Operating Mode & Structural Level:

The RD-TENG employs FCS mode with low tangential displacement and minimal friction, making it more durable than sliding-mode TENGs. We replace traditional inertial mass with an integrated elastic steel inner ring that functions as an oscillator, electrode, and structural support. Its elastic cantilever bridges enable soft contact, reducing impact, wear, and fatigue. Acting as a spring–mass system (Fig. S5, ESI†), it ensures gentle, durable operation over time.

Fig. S5. Vibration simulation of the vibrating steel sheet.

Added Fig. N16. Comparison of restoring forces for magnetic repulsion and traditional spring.

3. From the Perspective of Transmission Dynamics:

The durability test used a magnetic repulsion pendulum platform, chosen for its non-contact actuation capability (Note S8, Section 1.2, ESI†). The strong repulsive force near the stopper enables rebound without mechanical wear (Fig. N16, ESI†), avoiding fatigue issues common in spring systems. This ensures stable, long-term energy transfer to the RD-TENG, maintaining consistent mechanical response and repeatable electrical output.

4. Packaging Perspective:

To ensure long-term performance, RD-TENG features a fully enclosed structure with PLA casing, acrylic sealing plates, and industrial-grade waterproof sealant (e.g., Dow Corning 732). Layered ring gaskets further protect internal components from moisture and contaminants.

A 12,760-cycle durability test showed stable output with no significant degradation, confirming structural and functional reliability. We plan extended cycle and aging tests to further evaluate robustness. We thank the reviewer for highlighting this important issue, which is now addressed in the revised manuscript.

Revision in Main Manuscript:

Page 18, Line 456-463: The 1–3 electrode pair was operated at 1.4 Hz with a 10 mm amplitude for 12,760 continuous cycles, exhibiting almost no performance degradation (Supplementary Fig. 34). This exceptional stability is attributed to systematic optimizations in mechanism design, structural configuration, and encapsulation strategy. Specifically, the introduction of the three-electrode charge dispatch mechanism effectively mitigates electrostatic shielding and charge cancellation, thereby enhancing charge transfer continuity and output uniformity. The device further benefits from the intrinsically low-wear nature of the contact-separation mode, ensuring long-term mechanical reliability. Moreover, the magnetic repulsive pendulum provides non-contact and stable excitation, circumventing the fatigue issues typically associated with traditional elastic elements. Coupled with a high-density packaging design, the RD-TENG is able to operate reliably within a structurally stable and well-sealed environment.

4. Please explain the optimal condition at 2.4 Hz and 70 mm amplitude, how if the higher frequencies or amplitudes provide higher output performance?

Response: We thank the reviewer for highlighting the RD-TENG’s dynamic response and structural parameter matching, key to understanding its adaptability under varying excitations and advancing nonlinear TENG research. While we previously explanation of the “increase–then–decrease” output trend in Note S7 (formerly Note S3, ESI†), its brief mention and absence from the main text may have obscured its importance. We have now expanded Note S7 for greater clarity and systematic explanation.

1. Phase I: Output boost in initial frequency–amplitude increase

Under 0.6–2.4 Hz and 10–70 mm amplitude, stronger excitation improves output by increasing contact cycles, impact force, and separation range. Vibrating sheets maintain elastic deformation and mech-elect coupling, ensuring efficient energy conversion.

2. Phase II: Response saturation at optimal excitation point

At ~2.4 Hz and 70 mm amplitude, the excitation aligns with the vibrating sheet’s natural frequency, approaching sub-harmonic resonance. This ensures strong contact–separation, large vibrations, and stable charge generation—marking the system’s peak output condition.

3. Phase III: Performance degradation beyond optimal excitation

With further increases in excitation, the output decreases due to:

- 1) When excitation exceeds the natural frequency, delayed or incomplete contact-separation reduces energy coupling efficiency.
- 2) High-frequency excitation causes excessive rotor momentum, leading to slip, bounce, or adhesion at the interface and lowering per-cycle charge transfer.
- 3) Beyond structural limits, vibrating sheets may undergo torsion or off-axis deformation, resulting in irregular impacts, mechanical wear, charge leakage, and reduced output.
- 4) Though amplitude increases input energy, unchanged internal structural constraints cause mismatched motion, hindering effective mech-elect coupling.
- 5) At high frequencies, asynchronous responses among multiple vibrating sheets may introduce phase differences or signal cancellation, weakening total electrical output.

4. Potential pathways for further performance enhancement

The high output at 2.4 Hz and 70 mm amplitude results from optimal dynamic matching between excitation and RD-TENG structure. As discussed in Note S8. ESI†, RD-TENG is structurally scalable—vibrating sheets can be customized (e.g., number, length, or width of bridging arms) to tune their natural frequency and adapt to various excitation sources.

We sincerely thank the reviewer for this insightful question. Related results are provided in the revised Note S7, ESI†. Future work will explore vibration behavior under varied sheet

parameters and optimize response across broader frequency–amplitude conditions to enhance real-world adaptability.

Revision in *Supplementary Information*:

Summary section of Experiment 1 of Supplementary Note 7: ... (text omitted) This is because, at the initial stage of excitation, with constant amplitude, an increase in acceleration results in a rise in frequency. As the frequency increases, the contact-separation speed between the vibrating sheet and the fixing sheet accelerates, and the number of contact–separation cycles per unit time increases. Meanwhile, the enlarged amplitude enhances the contact pressure. The moving part gains higher velocity, thus higher momentum, which leads to more thorough contact with the fixing sheet and an increased output. These two effects work synergistically to enhance the efficiency of triboelectric charge generation and the rate of charge transfer, resulting in a steady improvement in output performance. As the excitation approaches 2.4 Hz / 70 mm, the frequency of external excitation nears the natural frequency of the vibrating sheet system, achieving complete and synchronous contact behavior and optimal coupling efficiency. However, as the frequency continues to increase, the structural response gradually lags behind the excitation rhythm, leading to non-ideal vibrations, slippage, and edge collisions. The moving part cannot separate and return in time during the short duration, resulting in a decrease in output. The overall energy conversion efficiency of the system thus declines. Furthermore, at high frequencies, multiple vibrating sheets are prone to phase mismatch and asynchronous responses, causing partial electrical output cancellation among different units and further weakening the overall performance. In summary, the output behavior of the RD-TENG is governed by the degree of matching between the excitation and structural dynamics, reflecting the inherently nonlinear coupling characteristics of the triboelectrification-electrostatic induction system. It should also be emphasized that this optimum point is not fixed but can be tuned by adjusting the parameters of the vibrating sheet to modify its natural frequency, thereby adapting to different external vibration environments and ensuring robust engineering scalability. ... (text omitted)

5. It looks like the RD-TENG operated under the nice ocean weather conditions, how about its efficiency under harsh ocean environments? Such as strong turbulence condition.

Response: We sincerely appreciate the reviewer’s attention to this practically significant engineering concern. This question not only pertains to the device’s adaptability during experimental validation but also directly addresses the feasibility and robustness of applying

TENGs in real-world ocean scenarios. We respond to your concern from two perspectives:

1. Sea conditions during testing and relevant application scenarios

1) Validation under mild-to-moderate sea states. In response to the reviewer's inquiry, real-time ocean data during testing (07:10, Nov 21, 2024; **Fig. S38**) showed: current 0.43 m/s, wave height ~0.8 m (max 1.1 m), and wind 29.13 km/h from the north. Though not extreme, these moderate ocean conditions involved wave, current, and wind disturbances. RD-TENG maintained stable operation throughout, with no structural or output failure (**Fig. S39**).

2) Device operation in calm sea states. The sea trial in "movie 9" was designed to assess RD-TENG performance under low-frequency, low-amplitude excitations during ebb tide, leveraging the magnetic repulsion pendulum (MRP) for scenario adaptability. Results confirmed that the RD-TENG, originally optimized for high-frequency input, effectively harvested weak wave energy and powered devices without external circuitry.

2. Energy conversion efficiency under extreme sea conditions.

We benchmarked the RD-TENG's energy conversion under real marine conditions against optimal lab 6-DOF excitation. In 1 minute, a 100 μF capacitor reached 4.06 V in lab tests, versus 1.09 V at sea, yielding 26.85% relative efficiency. This indicates a theoretical improvement margin of ~71.75% under stronger marine motion (**Fig. S40**). However, due to wave randomness and instability (**Fig. S41**), real-world efficiency is inherently limited compared to controlled lab inputs.

3. Protective strategies for extreme sea conditions.

We fully recognize that marine environments are highly unpredictable, especially under conditions such as storm surges, high-frequency turbulence, and compound wave interactions. These may cause large-amplitude pitching, overturning risks, or intensified structural impacts. In response, we have proposed two forward-looking pathways in Chapter 3 of Note S8:

1) Performance enhancement perspective: Under more intense excitation, if the structural integrity of the device is maintained, the effective contact frequency and rate of pendulum angle change could increase, theoretically boosting per-unit-time charge transfer and power output.

2) Stability and anti-disturbance design: To enhance RD-TENG's stability in harsh seas, we plan to integrate multiple degrees of freedom anti-roll systems, flexible constraints, high-strength materials, and smart buoy technologies. Future work will also involve wave tank testing and simulations to analyze dynamic responses across sea states.

We sincerely thank the reviewer for this highly valuable and practical question. Your comment highlights the importance of revealing operational boundaries in dynamic scenarios,

which is key to transitioning TENGs from lab research to real-world deployment. We have updated the manuscript and supplementary information with details on sea conditions, efficiency estimations, and system stability, and will continue to advance this work toward robust marine applications.

Revision in *Main Manuscript*:

Page 20, Line 494-497: ...Supplementary Fig. 38 shows real-time sea-state snapshots taken during data collection, illustrating mild-to-moderate sea conditions. Under these conditions, the RD-TENG and its buoyant platform maintained excellent stability without short circuits or structural damage, as demonstrated in Supplementary Fig. 39. Furthermore, the system exhibited outstanding wave-direction adaptability and wave-following response capability. ...

Page 20, Line 515-522: To evaluate the RD-TENG system's energy conversion efficiency under real-sea conditions, we compared capacitor charging performance (100 μ F, rectified electrode pair) against optimal laboratory conditions on the 6-DOF platform. After 1 min charging, the voltage reached 1.09 V at sea versus 4.06 V in the lab, achieving approximately 26.85% of laboratory efficiency (Supplementary Fig. 40). This result indicates substantial headroom (up to 71.75%) for performance enhancement under stronger marine excitation. However, due to the inherent randomness and irregularity of natural wave conditions, TENG triggering frequency and operation efficiency at sea were inevitably lower than under controlled lab settings (Supplementary Fig. 41).

Revision in *Supplementary Information*:

Fig. S38. Screenshot of meteorological and sea conditions at the beginning of data acquisition during real-sea testing.

Fig. S39. Real-sea output performance of the 1-3 electrode pair, showing no structural failure during the 14-minute testing period.

Fig. S40. Comparison of capacitive energy storage efficiency between real-sea condition and optimal condition on the six-degree-of-freedom simulation platform.

Fig. S41. Comparison of output data under real-sea conditions and laboratory wave tank mode 1 conditions

Note S8. Discussion on materials, engineering choices, scalability, limitations, and future strategies. Page 49: (As the content is lengthy, only a partial excerpt is provided here)

... (text omitted)

3. Potential limitations and future mitigation strategies.

Although the RD-TENG ... (text omitted) remain several limitations to be further addressed:

... (text omitted)

Moreover, the marine environment is inherently unpredictable, especially under extreme conditions such as storm surges, high-frequency turbulence, or the superposition of intense waves. In such scenarios, the MRP platform may experience severe swaying, overturning risks, or amplified structural shocks. In future work, we plan to enhance the anti-overturning capability and output stability of the RD-TENG under harsh sea states by integrating six-degree-of-freedom anti-tilt structures, flexible limiting frames, high-strength support materials, and intelligent buoy stabilization technologies. Simultaneously, we also intend to introduce numerical simulations and wave tank experiments to systematically investigate the dynamic response behaviors of the system under various sea conditions.

6. In this manuscript, the authors mention the technology roadmap of TENGs from water wave to marine IoT device applications, however, they are not demonstrated in the application part. Which specific IoT or sensing system you are targeting to show with this design?

Response: We sincerely thank the reviewer for raising this highly strategic and application-oriented question. Your comment prompted us to carefully reflect on the positioning of each stage in the energy chain within this work and its broader implications for future system integration and application expansion. We address your concern from the following two perspectives:

1. Guiding significance of the proposed technical roadmap.

Fig. 5a presents a full-chain technical roadmap for RD-TENG application in marine energy harvesting, covering wave capture, mechanical transmission, energy conversion, power management, and end-use deployment. This framework outlines four core stages:

- 1) Wave energy capture via a buoy–pendulum structure with asymmetric torque to enhance swing amplitude;
- 2) Mechanical transmission through a magnetic repulsion pendulum, converting wave motion into periodic TENG actuation;
- 3) Energy conversion by the RD-TENG, which employs a charge dispatch strategy to overcome stacking limitations and improve mech-elect efficiency;
- 4) Energy storage and output regulation via rectification modules to power external devices.

This roadmap reflects a systematic design strategy across structural, mechanical, electrical, and application levels. Beyond this work, the roadmap serves as a generalizable framework for marine energy harvesting systems, promoting modularity, scalability, and real-world deployment.

2. Adaptable Marine IoT Applications and Sensing Systems.

While this work does not yet integrate a specific IoT system, the RD-TENG’s demonstrated output, energy storage, load-driving capability, and stability under real-sea weak-wave conditions (e.g., powering a calculator) indicate strong potential for coupling with existing energy management circuits (e.g., buck converters, boost/switching modules). In testing, it reached 780 V peak output—sufficient to trigger gas discharge tubes and thyristors—enabling future integration with more complex marine IoT modules.

Our team is actively developing and exploring IoT application systems based on TENGs and blue energy, with a comprehensive focus on energy harvesting, sensing (e.g., temperature, humidity, wave height, wave velocity, ocean current), positioning, long-range wireless communication, and marine navigation. Significant achievements have already been made in blue energy harvesting (e.g., Adv. Energy Mater. 2025, 15, 2402781, Adv. Funct. Mater. 2024, 34, 2406775, Adv. Sci. 2022, 9 (35), 2204407), while steady progress is also being made in the other four pillars of marine IoT development. For instance, in Nano Energy 133 (2025) 110488, our team deployed a marine sensing system in the Beibu Gulf, achieving 1024 m remote data transmission. Adv. Sci. 2022, 9 (35), 2204407 proposed HM-TENG was utilized to power a radio frequency (RF) signal transmitter.

The RD-TENG was designed to demonstrate adaptability and platform-level feasibility in

complex marine environments. Current results validate its engineering potential. In the next phase, we aim to further advance the integrated application of RD-TENG in key areas such as marine wireless communication and remote localization, building a more comprehensive verification framework—ultimately accelerating the practical deployment and large-scale implementation of self-powered blue-energy IoT systems.

We sincerely thank the reviewer once again for raising this system-level perspective. We have refined the explanation and visual presentation of the technical roadmap in the main manuscript and in Fig. 5a. Additionally, we have emphasized at the end of the manuscript our plan for the next phase, which focuses on integrating the RD-TENG with marine IoT application modules.

Revision in Main Manuscript:

Revised Fig. 5a Technology roadmap of TENGs from water wave to marine IoT device applications.

Page 18-19, Line 483-492: ... The conversion of wave energy into electrical energy by TENG involves several stages: 1) **energy absorption**: transfer of wave energy to the device casing or wave-absorbing system, 2) **energy transfer**: from the wave-absorbing system to the TENG device, 3) **energy conversion**: the TENG device's output, and 4) **Energy management and supply**: output to energy storage or applications (Fig. 5a). ... thereby safeguarding overall wave-energy harvesting performance: 1) an unbalanced buoyant base captures wave energy; 2) the MRP transfers and amplifies the captured wave energy; 3) the RD-TENG device efficiently converts mechanical energy into electricity; and 4) the harvested energy is rectified, stored, and finally supplied to electronic devices.

Page 21, Line 528-531: ... and expecting to promote the development of the marine IoTs. In the next phase, we aim to further integrate the RD-TENG into critical marine applications such as wireless communications, remote positioning, and navigation guidance, establishing a more rigorous validation workflow. Ultimately, we strive to drive the practical and scalable

deployment of a self-powered blue-energy IoT network.

7. It is important to maintain synchronization and the stability of stacking multiple devices, how can you solve this issue?

Response: We sincerely thank the reviewer for raising the critical question regarding the “synchronization and stability of multi-unit stacked devices.” Your comment precisely highlights one of the key challenges faced when transitioning TENGs from single-unit designs to array-level deployment, and it touches upon a central technical issue addressed in our work related to system scalability. We will address your concern and resolve this issue from the following two aspects:

1. Current mitigation strategies

1) Electrical connection design: Self-inserting, weld-free electrical connection mechanism.

To address the instability and complexity of traditional cable connections, especially in humid or vibration-intensive conditions, we designed a modular plug-in electrical connection using conductive push pins and elastic conductive sponges (Fig. S21b, ESI†). This weld-free, self-aligned connection ensures stable conduction between electrode layers, significantly reducing output fluctuations and improving system integration efficiency and electrical stability.

2) Mechanical connection design: Rotating-clasp locking and standardized assembly.

To enhance structural stability and environmental resistance, we introduced a dual-locking mechanism combining a rotating-clasp device with standardized packaging barrels (Fig. S21a, ESI†). This design offers strong anti-loosening performance, resistance to vibration and impact, and supports repeated assembly, making it particularly suitable for modular array deployment in challenging environments like ocean waves.

Fig. S21a Details and physical image of the RD-TENG device array.

Fig. S21b Connection method of push-pin electrodes in the device array.

2. Future optimization directions

We acknowledge that, despite improved assembly ease and stability, our modular design faces

synchronization issues under high-frequency excitation. Inertial vibrations and nonlinear coupling from the linear-motor platform caused interlayer asynchronous responses, reducing output coherence. We appreciate the reviewer's concern and outline our optimization strategies:

1) Employ alternative excitation platforms or real-scenario validations. The current linear-motor test platform produces waveforms resembling square waves due to abrupt stops at high frequencies. Future studies will consider adopting wave-like periodic excitation platforms or practical environmental tests, such as wind-driven or ocean-wave-driven scenarios. Compared to reciprocating setups, floating platforms under wave excitation or periodic inclination exhibit smoother sinusoidal waveforms with minimal inertial discontinuities, likely providing more realistic evaluations of module coordination.

2) Incorporate flexible damping connection layers. Thin layers of flexible damping materials (such as silicone rings or EVA foam pads) will be integrated at module junctions. These layers maintain mechanical locking rigidity while absorbing inertial shocks at abrupt stops, thus reducing asynchronous responses between modules.

3) Introduce distributed limiting and buffering mechanisms. Minor limit structures or controllable damping supports will be strategically placed at the array perimeter or midpoint to enhance overall mechanical coordination.

4) Implement localized adaptive frequency-tuning designs. Future studies will experimentally adjust vibration-sheet parameters within modules, arranging them in arithmetic or geometric progressions to optimize local natural frequencies. Such tuning facilitates near-synchronous states under unified excitation, reducing output cancellations among modules.

We believe these strategies will notably improve synchronization and practical energy integration efficiency in TENG arrays. Future work will involve developing optimized testing platforms, refining array configurations, and further investigating the coupling between nonlinear phase disturbances and electrical outputs. Detailed discussions and optimizations are provided in the main text and Note S8, ESI†, Chapter 3.

Revision in *Main Manuscript*:

Page 13, Line 314-315: ... This causes tremors when the device reaches the end and reverses, affecting the synchronization of the vibrating sheets' movement (See the relevant analysis and proposed optimization strategies in Chapter 3 of Supplementary Note 8).

Revision in *Supplementary Information*: Chapter 3 of Supplementary Note 8:

Page 48-49: (As the content is lengthy, only a partial excerpt is provided here)

Note S8. Discussion on materials, engineering choices, scalability, limitations, and future strategies.

...

3. Potential limitations and future mitigation strategies.

... (text omitted)

Finally, although the rotating-clasp-device and contact push-pin connection strategies enhance the integration stability and electrical connectivity of RD-TENG arrays, ... (text omitted) Moving forward, we plan to introduce flexible damping layers, limiters, and local frequency tuning designs to improve mechanical coordination among modules... (text omitted)

Minor comments:

1. In fig. 1a-b-c, it is a need to explain clearly about the sheet 1-2-3, might be state them as fixing electrode 1, vibrating electrode 2, and fixing electrode 3?

Response: We sincerely thank the reviewer for the meticulous review and professional suggestion regarding the electrode labeling in Fig. 1a-b-c, which is highly valuable in enhancing both the technical accuracy of the illustration and the efficiency of reader comprehension. We fully recognize the rigor and constructiveness of your observation. Therefore, we have fully adopted your suggestion and have revised the labeling in Fig. 1a-c. We have also applied consistent modifications across all relevant illustrations and descriptions in both the main text and supplementary materials to ensure terminological consistency and clear correspondence between text and figures.

Revision in *Main Manuscript and Supplementary Information:*

The naming of each sheet has been revised in multiple sections of the main text, for example:

...When vibrating electrode 2 is stimulated by external mechanical energy, ... the non-coated vibrating electrode 2 oscillator contacts the left-coated fixing electrode 3, and the full-coated oscillator contacts the right-coated fixing electrode 1. Electrostatic induction causes the non-coated vibrating electrode 2 ... (text omitted)

Revised Figure 1a-c

Revised Fig. S3, ESI†

The unified figure annotations also include Fig. 2a-b, Fig. N1-N3, Fig. N5, Fig. N7-N8, etc.

2. In Figure 1e, it is a schematic of RD-TENG including 7 free-standing layer vertical contact-separation mode TENG (FCS-TENG). I suggest simplifying the schematic into 2 units at both end of RD-TENG, the middle can keep blank with 3 horizontal dots to simplify the schematic, easier for the reader.

Response: We sincerely thank the reviewer for the valuable suggestion regarding the schematic in Fig. 1e, which highlights your attention to visual clarity. As an overview diagram, Fig. 1e is critical in illustrating the RD-TENG's overall architecture and engineering design. Thus, ensuring its accuracy, clarity, and concise presentation is vital for the manuscript's logical coherence and readability.

We redrew Fig. 1e. Considering that a complete structural period of the RD-TENG consists of two independent FCS-layer units—namely, one with an uncoated vibrating sheet and the other with a fully coated vibrating sheet—we preserved one full RD-TENG structural period (comprising two FCS units) on the left end of the figure. On the right end, only the top capping layer is retained, while the central portion is abbreviated with an ellipsis ("...") to clearly indicate repetition and reduce visual complexity. Additionally, to address potential loss of structural detail from the necessary simplification, we have included a new schematic in the supplementary materials. This diagram provides a complete, expanded view of the RD-TENG with accurate dimensions, clearly illustrating the full stacking sequence of all seven internal

FCS-TENG units, detailed layer arrangements, and electrode annotations, enabling readers to thoroughly examine the device's internal structure.

Revision in Main Manuscript and Supplementary Information:

Page 6, Line 156: As shown in Fig. 1e, fixed sheets are placed on both sides of the vibrating sheet, ensuring contact separation to form an RD-TENG device. ... (text omitted) The full-model view and engineering layout are shown in Supplementary Fig. 6

Revised Fig. 1e Schematic diagram of the RD-TENG structure.

Added Fig. S6 (ESI†). Exploded isometric view and front view of the full-scale structural model of the RD-TENG with all components.

3. Is it duplicated figure between Figure 1 and Figure Note S2? The current value shown at the voltmeter are different, while the schematic showing the vibrating electrode 2 is at same position. Please clarify this issue.

Response: We thank the reviewer for this sharp and valuable observation regarding the partial redundancy between Figure 1 and the original Supplementary Figure Note S2, as well as the inconsistent ammeter readings. Your feedback is essential for improving the accuracy and consistency of our figure presentations.

We acknowledge partial redundancy between Figure 1 and the former Figure Note S2; both illustrate representative states of vibrating electrode 2. To address this, we have expanded the figure Note S2 (now Note S3, Fig. N5) to include all intermediate structural states throughout the full motion cycle for both the RD-TENG three-electrode configuration and the rattle-drum prototype, along with their conceptual and engineering evolution. The previously inconsistent current meter readings—despite identical physical states—were due to varying definitions of current direction and meter styles during drafting; all conventions have now been standardized to ensure clarity and consistency.

Revision in *Main Manuscript and Supplementary Information:*

Revised Figure 1a-c, e

Revised Figure N5 in Note S3. (Original Note S2 was merged into the additional Note S3)
The unified figure annotations also include Figure 2a-b, Fig. N1-2 in Note S1 and Fig. S3.

4. All the arrows in figures are not readable, please change it to be clearer and simple to the reader. In addition, in Figure 2a, the RD-TENG can be shown in cross-sectional view for better presentation.

Response: We sincerely thank the reviewer for the thoughtful suggestions regarding the clarity and readability of our figures. We fully recognize the importance of these issues in improving the visual communication of our work and have fully adopted your recommendations. Specifically, we removed redundant arrows from the figures, while retaining certain guide arrows with clear labels (e.g., Fig. 5a) to enhance interpretability. Furthermore, we revised Figure 2a(i) by adopting a cross-sectional schematic with appropriately thickened layers to more effectively illustrate the internal layout of a single structural cycle of the RD-TENG. Additional details of the push pin and latch positions are provided in Fig. 2a(ii). We also added detailed spatial orientation references and real model cross-sectional images in the supplementary information. This aims to help readers better grasp the core configuration and working principles of the device.

Revision in Main Manuscript and Supplementary Information:

Page 8, Line 205-210: ...Fig. 2a(i) shows a unit structural cycle cross-section image of the RD-TENG (For more details, see Supplementary Fig. 8), ... Fig. 2a(ii) and Supplementary Fig. 11 illustrate the left-side sectional view of the device, showing the staggered arrangement of three push pin holes and three latch holes arranged at 60° intervals along the circular periphery ...

Revised Fig. 2 Structural innovations of the RD-TENG. (a) Structural innovations of the RD-TENG. i) equivalent cross-sectional view of a structural cycle, ii) push pin positions, and latch positions. The three structural innovations: iii) Contact push pin electrode conduction

technology, iv) ... v) ...

Added Fig. S8 ESI† Cross-sectional schematic and orientation diagram of a single structural cycle of the RD-TENG.

Added Fig. S11 ESI† Structural details of push pin and latch alignment holes.

Reviewer #3 (Remarks to the Author):

The paper is very well written, The TENG design shows a fair amount of novelty and the results are interesting. Therefore, I believe that the paper could be worthy of publication after a few adjustments are suggested:

Response: We sincerely appreciate the reviewer's recognition and support for our work. Your comments highlighting the novelty of the proposed design, the value of the experimental results, and the overall quality of the manuscript are deeply encouraging for our team. The review and the following suggestions not only reflect a profound understanding of the field but also demonstrate a rigorous attitude toward enhancing scientific quality. We have carefully considered your feedback and will address each of the points in detail in the following responses. We believe these suggestions will further strengthen the scientific rigor and clarity of the manuscript, allowing us to better convey the innovations and engineering implications of this study.

1. I believe that there are statements in the introduction lack references, for example:

Line 43, Page 2: "However, the path to the commercialization of TENGs is fraught with stumbling blocks, including suboptimal output performance, limited mechanical lifespan, high fabrication costs, and limited adaptability to various environments."

Line 48, Page 2: "This disagreement stems not only from the significant variations in practical applications but also from the lack of a universally accepted, straightforward metric for evaluating the performance of different TENG structures. The introduction of triboelectric surface density (TSD) has, to some degree, provided a resolution to these controversies."

Response: We sincerely thank the reviewer for the careful reading and valuable feedback. Your comment regarding the lack of literature support for certain statements is particularly important, as it highlights the need to further strengthen the theoretical foundation and contextual relevance of our study. We apologize for this oversight. In response, we will carefully review all relevant sections, incorporate the necessary references where applicable.

We have supplemented the relevant sections of the manuscript with appropriate references to address the previous omissions. Regarding the statement "However, ... and limited adaptability to various environments," we have cited Chem. Eng. J. Adv. 2022, 9, 100237, which highlights that despite substantial progress in TENG development, the electrical output remains relatively low, and the key future challenges lie in further enhancing the output power and current, economically fabricating advanced TENGs, and designing systems capable of operating reliably under diverse real-world conditions. Our previous work (Adv. Mater.

Technol. 2025, e00184.) also emphasized the urgency of TENG commercialization and the multifaceted challenges involved.

In support of the discussion, “This disagreement stems not only ... provided a resolution to these controversies.” We have added citations as shown in the Table below:

Re.	Ref.	Content	Metrics
24	Nat. Commun. 2022, 13 (1)	proposed the concept of intrinsic surface charge density to quantify the theoretical output limits of different TENG materials.	SCD
25	Nat. Commun. 2021, 12 (1), 5470	introduced the surface instantaneous power density as a performance indicator.	SPD (surface power density)
26	Energy Environ. Sci. 2025, 18 (10), 4893–4904	evaluated TENG performance through surface charge density and energy density metrics.	SCD
27	Nat. Commun. 2015, 6 (1), 8376	established a set of standard figures of merit for quantifying TENG performance from both structural and material perspectives;	material and structural parameters
28	Research 2023, 6, 237	emphasized VPD as a key metric and explored how frequency enhancement mechanisms affect VPD.	Peak VPD (volume power density)
17	Adv. Mater. Technol. 2025, e00184.	clearly defined the physical meaning and scientific implications of TSD, and advocated its potential as a fair and universal metric for the structural evaluation of TENGs.	TSD

Revision in *Main Manuscript*:

Page 2, Line 44: However, the path to the commercialization of TENGs ... and limited adaptability to various environments.^{16,17}

Page 2, Line 50: This disagreement stems not only ... The introduction of triboelectric surface density (TSD) has,¹⁷ to some degree, provided a resolution to these controversies.^{24–28}

References

- Walden, R., Kumar, C., Mulvihill, D. M. & Pillai, S. C. Opportunities and challenges in triboelectric nanogenerator (TENG) based sustainable energy generation technologies: a mini-review. *Chem. Eng. J. Adv.* 9, 100237 (2022).
- Tang, W., Liu, G. & Wang, Z. L. Water-wave driven triboelectric nanogenerator networks: a decade of March in blue energy and beyond. *Adv. Mater. Technol.* 2025, e00184.
- Liu, D. et al. Standardized measurement of dielectric materials’ intrinsic triboelectric charge density through the suppression of air breakdown. *Nat. Commun.* 13, (2022).
- Wu, H., Wang, S., Wang, Z. & Zi, Y. Achieving ultrahigh instantaneous power density of 10 MW/m² by leveraging the opposite-charge-enhanced transistor-like triboelectric nanogenerator (OCT-TENG). *Nat. Commun.* 12, 5470 (2021).
- Guo, R. et al. Performance metrics of triboelectric nanogenerator toward record-high output energy density. *Energy Environ. Sci.* 18, 4893–4904 (2025).
- Zi, Y. et al. Standards and figure-of-merits for quantifying the performance of triboelectric nanogenerators. *Nat. Commun.* 6, 8376 (2015).
- Li, Z. et al. Standardized volume power density boost in frequency-up converted contact-separation mode triboelectric nanogenerators. *Research* 6, 237 (2023).

2. In figure 2b, it would be good to label that the fixed and vibrating sheets are electrodes (similar to the labels in figure in Note S2).

Response: We sincerely thank the reviewer for their meticulous examination of the figure details and for providing such valuable suggestions. This suggestion significantly enhances the structural logic and functional clarity of the illustrated model, helping readers better grasp the core components of the device design. We have fully adopted your recommendation and, in alignment with related comments from Reviewer #2, conducted a comprehensive optimization of the figure style and labeling system in Figure 2. Furthermore, to ensure consistency and professionalism throughout the manuscript, we have standardized all figure annotations referring to this structure in both the main text and supplementary materials. The updated content is presented below for your kind review. Once again, we extend our heartfelt thanks for your insightful feedback on the graphical representation, which has greatly contributed to improving the readability and professionalism of our manuscript.

Revision in Main Manuscript and Supplementary Information:

Revised Figure 2a-b

Revised Figure N5 in Note S3. (Original Note S2 was merged into the additional Note S3)
 The unified figure annotations also include Fig. 1a-c, e, Fig. N1–3, Fig. N7–8, Fig. S3, etc.

3. In page 13, Line 347. I would like to see a better explanation for why repulsive magnets were used in the structure of the magnetic repulsive pendulum (which I assume is acting like a spring).

Response: We sincerely appreciate the reviewer’s insightful question regarding the magnetic repulsive structure. We fully acknowledge and value your focus on the underlying mechanical mechanism. Indeed, in our study, the magnets within the repulsive system function similarly to traditional mechanical springs by providing a repeatable restoring force to drive pendulum oscillation under external excitation. However, compared to conventional spring-based systems, we deliberately chose a non-contact magnetic repulsive system as the energy restoring unit, owing to several structural and functional advantages:

1. **Exceptional force-displacement sensitivity and responsiveness.**
2. **Enhanced dynamic tunability and environmental adaptability.**
3. **Enhanced non-contact operation, low wear and fatigue, and high mechanical integration.**

These advantages align well with our design philosophy that emphasizes high sensitivity and adaptability in harvesting low-frequency, low-amplitude natural energy.

We have included in the *Supplementary Information* a detailed comparative discussion between magnetic repulsion and spring forces, covering theoretical modeling, force–

displacement visualizations, equivalent restoring stiffness theory, and other influencing factors, to further support our choice of magnets. We once again sincerely thank the reviewer for raising this critical point, which helped us clarify the scientific rationale behind our material selection and structural strategy.

Revision in *Main Manuscript*:

Page 15, Line 397-400: ... This repulsive force functions similarly to the restoring force of a spring; however, magnets are favored due to their superior response sensitivity, tunable dynamic characteristics, and mechanical and deployment advantages over traditional springs (Chapter 2 of Supplementary Note 8). ... (text omitted)

Revision in *Supplementary Information: Chapter 2 of Supplementary Note 8*:

Page 43-46: (As the content is lengthy, only a partial excerpt is provided here)

Note S8. Discussion on materials, engineering choices, scalability, limitations, and future strategies.

1. Selection of core materials and engineering design strategies:

(2) Magnet

In the scenario-adaptive extension strategy, magnets are employed as the restoring force source for oscillation, functioning analogously to mechanical springs by providing repeatable restoring forces under external excitation. However, ... (text omitted)

1) High-sensitivity response: Traditional springs rely on elastic deformation and follow

Hooke's Law, where the restoring force is proportional to displacement:

$$F_{spring}(r) = -kx \tag{1}$$

where k is the spring stiffness coefficient and x is the displacement, resulting in a linear restoring force. In contrast, the repulsive force between permanent magnets exhibits a distinctly nonlinear distance-dependent response. According to the magnetic dipole model, the repulsive force F between magnets varies with distance x approximately as:

$$F_{mag}(r) \propto \frac{A}{x^n} \tag{2}$$

A is the magnetic repulsion constant, influenced by the structural and material properties of the magnets. The exponent n is greater than 2, typically ranging between 2 and 4, depending on the magnet shape and boundary conditions. As shown in equation (2), the magnetic repulsive force increases rapidly as the distance between the magnets decreases, enabling a stronger restoring force even under small displacement perturbations. This results in a higher effective system stiffness during the initial response phase, which helps overcome startup inertia and

facilitates rapid activation at low frequencies.

To visually compare the magnitude and trend of restoring forces with a mechanical spring, we set $A = 0.5 \text{ N mm}^2$, $k = 1 \text{ N mm}^{-1}$, and $n = 2$, and plotted the force-displacement relationship in Fig. N16. In the low-displacement region (0-2.5 mm), the magnetic repulsion curve exhibits a steeper slope than the linear spring, indicating greater responsiveness and sensitivity to small excitations.

Fig. N16. Comparison of restoring forces for magnetic repulsion and traditional spring.

2) High dynamic tunability and environmental adaptability. The restoring stiffness k of traditional springs is typically fixed by their material properties and geometry, requiring physical replacement to alter performance. In contrast, the equivalent restoring stiffness k_{eff} of a magnetic repulsion system can be continuously tuned by adjusting the spacing, orientation, and size of the magnets. This is reflected in the repulsion constant A in equation (2) and Table N16. The simplified model is:

$$A \approx \frac{\mu_0}{4\pi} \frac{6 m_1 m_2}{1} \quad (3)$$

Table N16. Factors affecting the magnetic repulsion constant (A).

Parameter	Meaning
μ_0	Vacuum permeability, $\mu_0 = 4\pi \times 10^{-7} \text{ H m}^{-1}$
m_1, m_2	Effective magnetic moments (or approximated magnetic charges) of the two magnets
Geometric parameters	Magnet shape (e.g., cylinder, sphere), area, and alignment direction
Material properties	Type of permanent magnet (e.g., NdFeB, SmCo), magnetic energy product, etc.

Moreover, according to the natural frequency equation of a pendulum system in vibration theory:

... (text omitted)

3) Mechanical advantages. From an engineering standpoint, the magnetic repulsion system offers superior integration compatibility with our proposed floating oscillation platform. ...

(text omitted)

Point-to-Point Response to the Reviewer's Comments

(Comments in black, response in blue)

REVIEWERS' COMMENTS

Reviewer #1 (Remarks to the Author):

The authors have answered the comment from the last revision in detail. Responses from last revision provides a more robust and nuanced understanding of its contributions.

Response: We sincerely thank the reviewer for the thoughtful and supportive feedback. We greatly appreciate the reviewer's efforts and constructive input throughout the revision process, which have further strengthened the clarity and impact of our work.

Reviewer #2 (Remarks to the Author):

Response: We sincerely thank the reviewer for the recognition of our work. The reviewer's support is highly encouraging and motivates us to further advance our research in this field.

Reviewer #3 (Remarks to the Author):

I would like to note that there are instances were the word "sheet" was misspelled as "sheep" (lines 120 and 135). Other than that, I believe that the revised version of this manuscript addressed the concerns raised in my previous revision, and I therefor can recommend this manuscript for publication.

Response: We sincerely thank the reviewer for the careful reading and constructive feedback. We apologize for the minor spelling error where "sheet" was mistakenly written as "sheep". The mistake has been corrected, and we have carefully rechecked the entire manuscript to avoid similar oversights. We are grateful for the reviewer's recognition and recommendation for acceptance, which is of great importance to the dissemination of this work.

Revision in Main Manuscript:

Original line 120: ... the non-coated **sheet** to the full-coated **sheet**, ...

Original line 135: ...the vicinity of fixing electrode 1 to fixing electrode 3. **Sheets** 2 and 4, ...